# Point4bit: Post Training 4-bit Quantization for Point Cloud 3D Detection

**Jianyu Wang**[1] , **Yu Wang**[1] , **Shengjie Zhao**[1*] , **Sifan Zhou**[2*]

[1]Tongji University
[2]Carnegie Mellon University

## Abstract

Voxel-based 3D object detectors have achieved remarkable performance in point cloud perception, yet their high computational and memory demands pose significant challenges for deployment on resource-constrained edge devices. Post-training quantization (PTQ) provides a practical means to compress models and accelerate inference; however, existing PTQ methods for point cloud detection are typically limited to INT8 and lack support for lower-bit formats such as INT4, which restricts their deployment potential. In this paper, we present Point4bit, the first general 4-bit PTQ framework tailored for voxel-based 3D object detectors. To tackle challenges in low-bit quantization, we propose two key techniques: **(1) Foreground-aware Piecewise Activation Quantization (FA-PAQ)**, which leverages foreground structural cues to improve the quantization of sparse activations; and **(2) Gradient-guided Key Weight Quantization (G-KWQ)**, which preserves task-critical weights through gradient-based analysis to reduce quantization-induced degradation. Extensive experiments demonstrate that Point4bit achieves INT4 quantization with minimal accuracy loss with less than 1.5% accuracy drop. Moreover, we validate its generalization ability on point cloud classification and segmentation tasks, demonstrating broad applicability. Our method further advances the bit-width limitation of point cloud quantization to 4 bits, demonstrating strong potential for efficient deployment on resource-constrained edge devices.

## 1   Introduction

3D object detection [25, 34] plays a crucial role in autonomous driving by enabling accurate environmental perception. Compared with camera-based methods, LiDAR offers robust and lighting-invariant geometric information, making it a popular choice for reliable 3D perception [37, 33, 44, 45]. Among various approaches, voxel-based 3D detectors [18, 47, 6] are particularly popular due to their ability to convert irregular point clouds into structured voxel grids, enabling efficient feature extraction via 2D convolutions. These methods are typically built on dense or sparse backbones [56, 46, 10, 52], aiming to improve accuracy and speed. However, real-time deployment remains challenging on edge devices. For instance, VoxelNeXt [6] achieves 26.9 FPS on an NVIDIA RTX 3090, which is impractical for in-vehicle deployment. These real-world constraints underline the urgent need to reduce the computational cost and memory footprint of voxel-based detectors, and to strike a better balance between accuracy and efficiency for large-scale, real-world applications.

Quantization is a promising solution to reduce model size and latency by converting floating-point (FP) operations into low-bit formats (e.g., INT8 or INT4) in different application [14, 43, 16, 53, 15]. Compared to quantization-aware training (QAT), post-training quantization (PTQ) is more practical, requiring no retraining and only minimal unlabeled data. A pioneering PTQ method for point cloud object detection is Lidar-PTQ [51], which systematically analyzes the reason for quantization error and

---

*Corresponding author: shengjiezhao@tongji.edu.cn, sifanjay@gmail.com

39th Conference on Neural Information Processing Systems (NeurIPS 2025).

proposes a compensation mechanism based on quantization-aware distribution alignment. While this method demonstrates strong practical value, it still has notable limitations: **(1) Limited robustness in ultra-low-bit settings**. While supporting INT8, naively lowering to INT4, already available on modern hardware [30], causes severe degradation, as uniform treatment of weights/activations breaks down at 4-bit where errors are amplified; **(2) Distillation-based optimization and time-consuming**. Its calibration and optimization procedures involve multiple training-style steps and are relatively complex, often requiring several hours to complete. This significantly hinders rapid deployment and iterative prototyping on edge devices; **(3) Limited generalization ability**. LiDAR-PTQ is primarily designed for 3D object detection models, and its effectiveness on other LiDAR-based perception tasks beyond 3D object detection remains to be explored.

To address the limitations of existing PTQ methods in terms of low-bit support, deployment efficiency, and task generalization, we propose a generic 4-bit PTQ framework for point cloud perception, named Point4bit. The framework consists of two key components: the Foreground-aware Piecewise Activation Quantization (FA-PAQ) and the Gradient-guided Key Weight Quantization (G-KWQ). The FA-PAQ leverages the inherent sparsity and geometric structure of point clouds to preserve task-critical features, especially those of foreground points. The G-KWQ module leverages gradient sensitivity analysis to identify task-critical weights and applies high-fidelity quantization to preserve model performance, even under extremely low-bit settings such as INT4.

Point4bit demonstrates strong quantization performance on voxel-based 3D detectors under INT4 settings, with minimal accuracy degradation. It also demonstrates robust generalization to classification and segmentation tasks, indicating broad applicability. In addition, Point4bit enables efficient deployment with minimal calibration overhead, making it a practical solution for resource-constrained edge scenarios. The main contributions of this paper are:

- We propose the Foreground-aware Piecewise Activation Quantization (FA-PAQ) design to address the issue of the challenging problem of sparse activation quantization in point cloud detection. By leveraging the structural information of foreground regions, FA-PAQ effectively mitigates performance degradation during low-bit settings.

- We introduce Gradient-guided Key Weight Quantization (G-KWQ), a gradient-guided weight quantization strategy that adaptively identifies and prioritizes task-critical weights. It effectively alleviates quantization-induced error propagation in low-bit regimes, thereby improving quantization performance.

- We develop Point4bit, the first general 4-bit PTQ framework for point cloud object detection. It achieves a favorable trade-off between compression ratio and accuracy. Experiments show that Point4bit consistently outperforms existing methods under both INT8 and INT4 settings, demonstrating strong robustness under low-bit quantization, superior cross-task generalization, and high deployment potential.

## 2 Preliminary

**Uniform Affine Quantization.** In quantization, uniform affine quantization maps FP inputs to a fixed integer range using three key parameters: the scale factor $s$, zero-point $z$, and bit-width $b$. Given an FP input $\mathbf{x}$ (weights or activations), the quantization and de-quantization process can be described by Eq. (1), where $\lfloor \cdot \rceil$ denotes the round-to-nearest operation (RTN), and $\mathrm{clamp}(\cdot)$ restricts the result to the valid integer range:

$$\mathbf{x}_{\mathrm{int}} = \mathrm{clamp}\left(\left\lfloor \frac{\mathbf{x}}{s} \right\rceil + z;\, 0,\, 2^b - 1\right), \widehat{\mathbf{x}} = s\left(\mathbf{x}_{\mathrm{int}} - z\right). \tag{1}$$

where $x_{int}$ represents the quantized integer value, $\hat{x}$ is the de-quantized FP value with an error that is introduced during the quantization process. Based on Eq. (1), the FP range of the quantization grid is defined as $[q_{\min},\, q_{\max}] = [-sz,\, s(2^b - 1 - z)]$. Input values $\mathbf{x}$ falling outside this range are clipped to the nearest boundary, resulting in clipping error. Increasing the scale factor $s$ can reduce clipping by expanding the representable range, but at the cost of increased rounding error, which is bounded within $\left[-\frac{1}{2}s,\, \frac{1}{2}s\right]$. To balance these two sources of error, modern quantization frameworks commonly adopt MSE-based calibration [2, 28], which searches for the optimal $(q_{\min}, q_{\max})$ that minimizes the Mean Squared Error (MSE) between the original tensor $\mathbf{x}$ and its de-quantized approximation $\widehat{\mathbf{x}}$.

# 3 Method

In this section, we present Point4bit, which consists of two key components: a foreground-aware activation quantization strategy and a gradient-guided weight quantization method. Inspired by LiDAR-PTQ [51], we enhance activation quantization by emphasizing foreground regions to better preserve task-relevant features. We further extend task-awareness to weight quantization by leveraging gradient information to retain weights most critical to the detection loss. Together, these components enhance the robustness and accuracy of quantized models under ultra-low-bit settings.

## 3.1 Foreground-aware Piecewise Activation Quantization

As pointed out in LiDAR-PTQ, current point cloud detection models are highly sensitive to activation quantization. To address the unique challenges posed by activation quantization in 3D object detection, we propose a novel method called Foreground-aware Piecewise Activation Quantization (FA-PAQ), which leverages the inherent sparsity and geometric structure of point clouds. This method aims to preserve task-critical features during quantization, especially those associated with foreground points, which are crucial for accurate 3D object detection.

FA-PAQ consists of two components: Adaptive Foreground Recognition and Foreground-aware Piecewise Activation Quantization. The former selects high-importance regions based on the activation map, while the latter applies finer-grained quantization in these regions to minimize activation information loss and maintain detection accuracy.

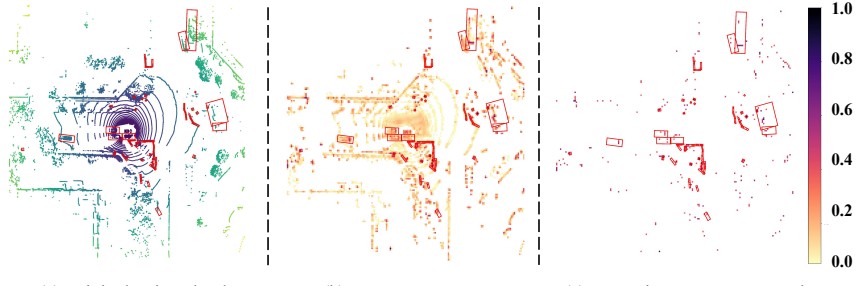

| (a) Original Point Cloud | (b) BEV Feature Map | (c) Top-K in Non-Empty Voxel |

Figure 1: Visualization of activation strength on the BEV feature map based on CP-Voxel [47]. Red boxes represent the regions of GT bounding boxes. Top-K denotes the top K non-empty voxels selected based on average activation responses.

**Adaptive Foreground Recognition.** LiDAR-PTQ reveals that the boundary values of activations significantly impact quantization performance. This insight motivates us to investigate which regions in LiDAR-based point cloud detection are more critical to maintain under quantization. To this end, we visualize both the raw input point cloud and the BEV feature maps in the network, as shown in Fig. 1. Surprisingly, when computing the mean activation over the channel dimension and sorting the spatial locations in descending order, we find that the Top-K non-empty voxels are almost entirely located within ground truth (GT) bounding boxes.

This observation implies that foreground regions tend to have larger activation magnitudes, while background activations are often close to zero. Therefore, activation magnitude can effectively and automatically identify foreground points in the BEV feature map under an unsupervised setting. These foreground responses are highly relevant to the detection task [22].

Formally, let the activation feature at $\ell$ layer be denoted as $X^\ell \in \mathbb{R}^{L^\ell \times C^\ell}$, where $L^\ell$ is the number of spatial locations, and $C^\ell$ is the number of channels. Let $\mathcal{V}^\ell \subseteq \{1, \ldots, L^\ell\}$ denote the set of non-empty voxel indices at layer $\ell$. For each location $i \in \mathcal{V}^\ell$, we compute the average activation across channels, and select the top $K = m_1 \cdot |\mathcal{V}^\ell|$ positions with the highest average activations as foreground candidates (Eq. (2)):

$$X^{\ell,fg} = \text{Top}_K \left( X_i^{\ell*} \right), X_i^{\ell*} = \frac{1}{C^\ell} \sum_{j=0}^{C^\ell - 1} X_{i,j}^\ell, \tag{2}$$

where $\text{Top}_k(\cdot)$ denotes selecting the top $k$ locations with the highest average activation. The resulting foreground set $FG$ is considered to contain the most critical structural information and will serve as the foundation for the subsequent piecewise quantization process.

As shown in Fig. 1(c), the selected Top-K activation locations strongly align with GT bounding boxes, validating the effectiveness of our unsupervised foreground extraction strategy. By incorporating this foreground-aware mechanism, we are able to apply finer-grained quantization in crucial regions, preserving geometry-sensitive activations while maintaining overall model efficiency.

**Piece-wise Activation Quantization** After identifying foreground features and analyzing their distribution, we propose a piecewise activation quantization strategy that determines scale factors for multiple intervals based on the distribution of foreground non-empty voxels in the BEV feature map. A straightforward approach is to uniformly divide the activation range; however, such uniform partitioning neglects the intensity distribution of critical features. In contrast, leveraging the cumulative distribution function (CDF) allows us to better account for the underlying distribution of feature intensities, leading to a more adaptive and data-aware quantization scheme (see Appendix for details). For background features, we adopt the conventional quantization strategy. This method ensures that the quantization process is well aligned with the underlying data distribution. In particular, the proposed approach is designed to enhance point cloud detection performance by preserving critical semantic information in foreground regions.

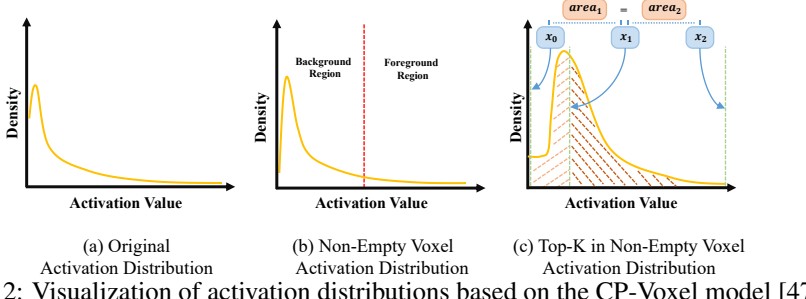

(a) Original
Activation Distribution

(b) Non-Empty Voxel
Activation Distribution

(c) Top-K in Non-Empty Voxel
Activation Distribution

Figure 2: Visualization of activation distributions based on the CP-Voxel model [47].

Foreground voxels in a BEV feature map typically correspond to semantically significant objects, such as vehicles or pedestrians, and are characterized by higher feature values or densities. We represent these feature values as a random variable $X$, whose statistical distribution is inferred from a calibration dataset. This distribution encapsulates the properties of foreground voxels within the feature space, providing the foundation for subsequent quantization steps.

*Step 1: CDF Estimation.* The cumulative distribution function $F_X(x)$ is defined as the probability that the feature value $X$ is less than or equal to a specific value $x$: $F_X(x) = P(X \leq x)$. To empirically estimate the foreground CDF $F_X(x)$, we collect a set of foreground feature values denoted as $X^{fg} = \{x_1, x_2, \ldots, x_n\}$. These values correspond to the activation intensities of the most semantically meaningful (foreground) regions. As illustrated in Fig. 2, the foreground activations exhibit a skewed distribution, with higher densities concentrated in specific value ranges. We sort the collected values in ascending order as $x_{(1)} \leq x_{(2)} \leq \cdots \leq x_{(n)}$. The empirical CDF is then approximated as shown in Eq. (3), where $\mathbf{1}(\cdot)$ denotes the indicator function, returning 1 if $x_{(i)} \leq x$ and 0 otherwise:

$$F_X(x) \approx \begin{cases} 0, & x < x_{(1)} \text{ or } x > x_{(n)} \\ \frac{1}{n} \sum_{i=1}^{n} \mathbf{1}(x_{(i)} \leq x), & x_{(1)} \leq x \leq x_{(n)}. \end{cases} \tag{3}$$

*Step 2: Interval Division.* To determine quantization intervals adaptively, we partition the feature value range into $m$ intervals, each containing an equal cumulative probability of $\frac{1}{m}$. The boundary points $p_k$ for $k = 1, 2, \ldots, m-1$ are selected based on the Eq. (4). The resulting intervals are defined as $[p_0, p_1], [p_1, p_2], \ldots, [p_{m-1}, p_m]$, where $p_0 = \min(X^{fg})$ and $p_m = \max(X^{fg})$ represent the minimum and maximum feature values, respectively. This equal-probability partitioning ensures that the quantization intervals reflect the data distribution (Fig. 2(c)).

$$F_X(p_k) = \frac{k}{m}. \tag{4}$$

*Step 3: Activation Quantization.* For each quantization interval $[p_{k-1}, p_k]$, we compute the quantization scale $s_k^{fg}$ for $b$-bit quantization (i.e., $2^b$ quantization levels), as shown in Eq. (5):

$$s_k^{fg} = \frac{p_k - p_{k-1}}{2^b}. \tag{5}$$

Based on the general quantization and de-quantization functions defined in Eq. (1), our piece-wise strategy derives a specific quantization rule for each foreground interval, as formulated in Eq. (6):

$$x^{\hat{f}g} = p_{k-1} + s_k^{fg} \cdot \text{clamp}\left(\left\lfloor \frac{x^{fg} - p_{k-1}}{s_k^{fg}} \right\rfloor ; 0, 2^b - 1 \right), x^{fg} \in [p_{k-1}, p_k]. \tag{6}$$

## 3.2 Gradient-guided Key Weight Quantization.

To identify weights that are critical to task performance in point cloud object detection, we propose a gradient-guided key weight quantization (G-KWQ). This method is motivated by the observation that 3D object detection heavily relies on geometric structure modeling. In sparse point cloud scenarios, structural cues such as object boundaries and shape information are particularly crucial for accurate detection. Therefore, we design a gradient-based weight sensitivity evaluation mechanism tailored for 3D detection tasks, which guides the preservation of high-fidelity representations for important weights during quantization.

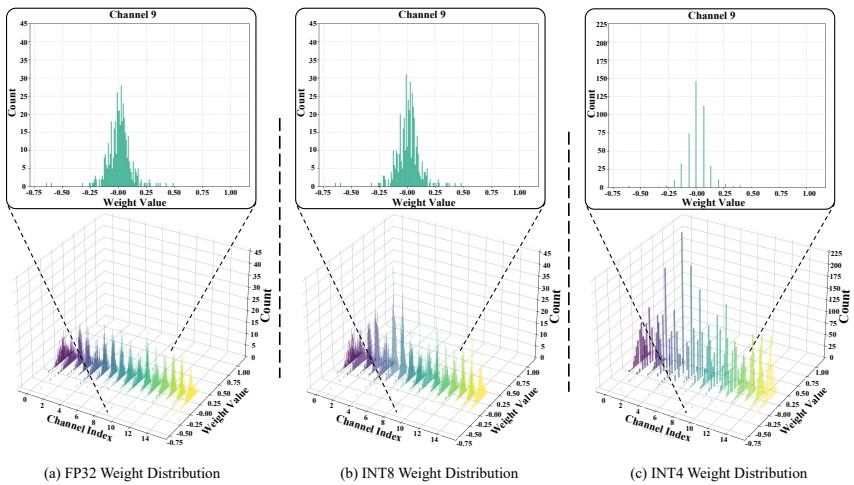

(a) FP32 Weight Distribution  (b) INT8 Weight Distribution  (c) INT4 Weight Distribution

Figure 3: Visualization of weight distributions based on the CP-Voxel model [47].

**Gradient-Based Weight Sensitivity Evaluation.** The magnitude of the gradient of a weight with respect to the loss function can indicate its task sensitivity. Specifically, the sensitivity of a weight $w_i$ is measured by the gradient magnitude of the total task loss $\mathcal{L}_{\text{task}}$, which typically consists of a Focal Loss for classification and a Smooth-$L_1$ Loss for regression. Given that our quantization is conducted in a per-channel manner, we compute the average gradient magnitude across input channels for each output channel, as shown in Eq. (7):

$$\alpha_j^\ell = \frac{1}{M^\ell} \sum_{j=0}^{M^\ell-1} \left| \frac{\partial \mathcal{L}_{task}}{\partial W_{i,j}^\ell} \right|. \tag{7}$$

Here, $\alpha_j^\ell$ denotes the aggregated task sensitivity of the $j$-th channel in the weight tensor of layer $\ell$, where a larger value indicates a potentially greater impact on overall model performance. $M^\ell$ represents the total number of weights in layer $\ell$. This sensitivity scoring strategy is particularly suitable for 3D object detection tasks that rely on structural information, as high-sensitivity weights often encode critical geometric features, such as object boundaries or dense spatial regions, which play an important role in determining the mean Average Precision (mAP).

**Rounding Error in Low-Bit Quantization.** As shown in Fig. 3, we observe that as the bit-width decreases, the distribution of quantized weights becomes increasingly sparse. This trend reflects a growing discrepancy between the original full-precision weights and their low-bit representations, which is primarily caused by rounding errors. The sparsity arises due to the limited set of representable values in low-bit quantization, leading to coarse-grained mappings and aggressive rounding, especially for weights near zero or quantization thresholds. Such increased rounding error may significantly degrade the network's capacity to retain subtle but task-relevant features. This observation highlights the necessity of explicitly mitigating rounding error for task-sensitive weights under low-bit settings.

**Key Weight Identification and Differentiated Quantization Process.** Based on the above analysis, we propose the gradient-guided quantization pipeline:

*Step 1: Gradient Extraction.* Perform a forward and backward pass on a calibration dataset to compute the per-channel gradient sensitivity using Eq. (7).

*Step 2: Sensitivity Scoring.* Use the gradient magnitude $\alpha_i^\ell$ (or its square) as a sensitivity score to measure each channel's importance to task performance.

*Step 3: Key Channel Selection.* Rank channels by $\alpha_i^\ell$ in descending order and select the top $m_2\%$ as task-critical channels for special treatment in quantization.

*Step 4: Differentiated Quantization.* Introduce a rounding error penalty term into the quantization loss to enforce high-fidelity quantization for important weights. The total quantization loss is defined as Eq. (8), where $\mathcal{L}_{\text{RE},j}$ denotes the rounding error and $S_j^w$ is the quantization step size for channel $j$:

$$\mathcal{L}_{\text{quant}} = \mathcal{L}_{\text{MSE}} + \lambda \sum_j \alpha_j \cdot \mathcal{L}_{\text{RE},j}, \mathcal{L}_{\text{RE},j} = \left( \frac{W_j}{S_j^w} - \left\lfloor \frac{W_j}{S_j^w} \right\rfloor \right)^2 \tag{8}$$

**Task-Specific Benefits.** Without introducing additional model complexity, our method adaptively identifies task-relevant weights using gradient information and applies differentiated quantization strategies to preserve their fidelity. Experimental results show that under low-bit settings (e.g., INT4), our method effectively mitigates performance degradation and demonstrates superior robustness and task adaptability in 3D detection scenarios.

### 3.3 Point4bit Algorithm

Point4bit is a fully post-training quantization method that operates without the need for labeled data or model training, making it particularly well-suited for deployment in privacy-sensitive or resource-constrained environments. The overall procedure of Point4bit is summarized in Algorithm 1.

---

**Algorithm 1** Point4bit Quantization

---

**Input**: Pretrained FP model with $N$ layers; Calibration dataset $D^c$.
**Output**: quantization parameters of both activation and weight in network, i.e., weight scale $s^w$, weight zero-point $z_w$, activation scale $\{s_1^{fg}, ..., s_m^{fg}, s^{bg}\}$.
 1: **for** $L_n = \{\ell_i | i = 1, 2, ...N\}$ **do**
 2:    Run a forward and backward pass on the calibration dataset $D^c$ to compute per-channel gradient sensitivity $\alpha_j^{\ell_i}$ using Eq. (7);
 3:    Optimize the weight quantization parameters $s^w$ and $z^w$ by minimizing the MSE loss defined in Eq. (8) using grid search algorithm.
 4: **end for**
 5: **for** $L_n = \{\ell_i | i = 1, 2, ...N\}$ **do**
 6:    Input $D^c$ to FP network and obtain the output feature $X_{\ell_i}$ at layer $\ell_i$;
 7:    Identify the foreground and background activation values, $X^{fg}$ and $X^{bg}$, according to Eq. (2);
 8:    Optimize the background activation quantization parameters $s^{bg}$ based on $X^{bg}$ by minimizing the MSE loss using grid search algorithm;
 9:    Determine the interval boundaries $\{p_0, p_1, ..., p_m\}$ based on the empirical cumulative distribution function as defined in Eq. (3) and Eq. (4);
10:    Optimize the foreground activation scales $\{s_1^{fg}, ..., s_m^{fg}\}$ using Eq. (5).
11: **end for**

---

## 4 Experiments

**Datasets.** We evaluate the effectiveness of the Point4bit framework primarily on the large-scale autonomous driving dataset nuScenes [4] for the 3D object detection task. The evaluation metrics for 3D detection are mean Average Precision (mAP) and the nuScenes Detection Score (NDS). To further assess its generalizability across different tasks, we conduct additional experiments on 3D object classification and semantic segmentation. For classification, we adopt the ModelNet40 [42] and ScanObjectNN [39] datasets, which are widely used as benchmarks in this domain, and evaluate performance using Overall Accuracy (OA) and mean Class Accuracy (mAcc). For semantic segmentation, evaluations are performed on the real-world LiDAR dataset SemanticKITTI [3], using mean Intersection-over-Union (mIoU) as the evaluation metric.

Table 1: CP-Voxel [47] Quantization results on nuScenes *val* set. GS: grid search strategy.

| Methods | Bits(W/A) | mAP | NDS | Car | Truck | C.V. | Bus | Trailer | Barrier | Motor. | Bike | Ped. | T.C. |
|---|---|---|---|---|---|---|---|---|---|---|---|---|---|
| Full Prec. | 32/32 | 58.45 | 66.22 | 84.81 | 57.09 | 15.88 | 70.08 | 37.66 | 67.37 | 57.39 | 39.35 | 84.99 | 69.92 |
| RTN [28] | 8/8 | 58.22 | 66.08 | 84.72 | 56.96 | 15.23 | 70.14 | 37.19 | 67.15 | 56.97 | 38.88 | 84.97 | 69.91 |
| RTN+GS [28] | 8/8 | 58.30 | 66.09 | 84.78 | 56.99 | 15.78 | **70.17** | 37.25 | 67.31 | 57.24 | 38.71 | 84.91 | 69.79 |
| PD-Quant [21] | 8/8 | 58.06 | 65.91 | 84.78 | 56.70 | 15.53 | 70.10 | 36.03 | 67.30 | 56.26 | 39.11 | 84.85 | 69.91 |
| QDrop [41] | 8/8 | 57.98 | 65.75 | 84.66 | 56.01 | 15.23 | 69.97 | 36.81 | 67.02 | 56.66 | 38.79 | 84.92 | 69.73 |
| LiDAR-PTQ [51] | 8/8 | 58.34 | 66.11 | **84.81** | 56.77 | 15.60 | 70.07 | **37.65** | 67.33 | 57.39 | 39.15 | 84.97 | 69.68 |
| Ours | 8/8 | **58.48** | **66.21** | 84.80 | 57.08 | **16.12** | 70.14 | 37.52 | **67.39** | 57.63 | 39.17 | 85.00 | 69.96 |
| RTN [28] | 4/8 | 55.80 | 64.30 | 82.30 | 54.40 | 18.16 | 66.35 | 36.78 | 66.53 | 51.96 | 31.50 | 82.78 | 67.22 |
| RTN+GS [28] | 4/8 | 56.63 | 64.67 | 84.07 | 55.86 | 12.41 | 67.56 | 36.29 | 66.35 | 54.55 | 36.92 | 84.56 | 67.66 |
| Ours | 4/8 | **57.46** | **65.34** | 84.15 | 55.83 | 17.77 | **68.41** | 36.37 | 66.89 | 56.37 | 36.27 | 84.25 | 68.23 |
| RTN [28] | 4/4 | 29.46 | 46.43 | 58.22 | 30.21 | 0.02 | 41.75 | 11.36 | 39.66 | 8.98 | 1.78 | 59.27 | 43.34 |
| RTN+GS [28] | 4/4 | 39.40 | 53.40 | 77.51 | 42.58 | 11.17 | 52.29 | 22.67 | 61.45 | 18.23 | 4.62 | 64.05 | 39.36 |
| Ours | 4/4 | **56.97** | **64.88** | **83.68** | **55.17** | **18.06** | **67.55** | **35.38** | **66.51** | **55.89** | **35.55** | **83.98** | **67.93** |

Table 2: Quantization results on the nuScenes *val* set based on VoxelNeXt [6].

| Methods | Bits(W/A) | mAP | NDS | Car | Truck | C.V. | Bus | Trailer | Barrier | Motor. | Bike | Ped. | T.C. |
|---|---|---|---|---|---|---|---|---|---|---|---|---|---|
| Full Prec. | 32/32 | 60.53 | 66.64 | 83.87 | 55.52 | 21.04 | 70.49 | 38.06 | 69.38 | 62.80 | 49.98 | 84.58 | 69.38 |
| RTN [28] | 8/8 | 42.48 | 44.67 | 77.49 | 43.41 | 9.81 | 46.74 | 21.44 | 47.45 | 31.75 | 12.58 | 76.06 | 58.05 |
| RTN+GS [28] | 8/8 | 60.40 | 66.58 | 83.85 | 55.61 | **21.05** | 70.40 | 37.93 | 69.25 | 62.61 | 49.68 | 84.46 | 69.08 |
| Ours | 8/8 | **60.50** | **66.63** | 83.87 | 55.61 | 21.03 | 70.46 | **38.06** | 69.25 | 62.87 | 49.93 | 84.55 | 69.29 |
| RTN [28] | 4/8 | 56.76 | 65.12 | **84.35** | **55.91** | 12.31 | 67.50 | 36.30 | 66.16 | 55.39 | 37.26 | **84.44** | **67.93** |
| RTN+GS [28] | 4/8 | 58.54 | 65.31 | 81.21 | 54.18 | 20.26 | 68.24 | 37.73 | 69.07 | 58.21 | 46.83 | 83.02 | 66.55 |
| Ours | 4/8 | **59.27** | **65.54** | 82.16 | 54.43 | **20.71** | **69.78** | **36.88** | **69.38** | **61.19** | **47.30** | 83.10 | 67.74 |
| RTN [28] | 4/4 | 20.72 | 29.24 | 54.58 | 19.68 | 0.0 | 26.65 | 5.60 | 09.08 | 0.01 | 0.0 | 57.30 | 34.24 |
| RTN+GS [28] | 4/4 | 52.71 | 61.36 | 79.24 | 48.44 | 18.40 | 60.60 | 31.39 | 66.88 | 52.34 | 38.46 | 79.13 | 52.13 |
| Ours | 4/4 | **58.97** | **65.20** | **81.89** | **54.03** | **20.39** | **69.76** | **36.98** | **69.07** | **60.76** | **46.90** | **82.68** | **67.15** |

**Implementation Details.** For the nuScenes dataset, we randomly sample 256 point cloud frames from the *train* set as calibration data, accounting for only **0.91%** of the total training frames (256/28,130). Similarly, 32 shapes are selected from the ModelNet40 *train* set (32/9,843, **0.32%**), 32 shapes are selected from the ScanObjectNN *train* set (128/11609, **0.01%**), and 128 frames are sampled from the SemanticKITTI *train* set (128/19,130, **0.67%**) for calibration. In all experiments, the first and last layers of the network are kept in full precision to preserve input fidelity and maintain output accuracy. Additional implementation details are provided in the supplementary material. We execute all experiments on a single Nvidia Tesla V100 GPU.

## 4.1 Overall Results

**Quantization Results on Point Cloud Detection.** We evaluate our Point4bit on the nuScenes *val* set for the 3D object detection task. We first adopt CenterPoint-Voxel [47] (CP-Voxel) as the full-precision baseline due to it is a representative voxel-based detector that strikes a good balance between accuracy and efficiency, and has been widely adopted in both academia and industry. As shown in Tab. 1, under the standard W8A8, Point4bit achieves 58.48% mAP and 66.21% NDS, which is nearly lossless compared to the FP model (58.45% mAP, 66.22% NDS), and outperforms existing PTQ methods including LiDAR-PTQ [51], QDrop [41], and PD-Quant [21]. To ensure compatibility with edge deployment, we also evaluate the mixed-precision W4A8 setting, which corresponds to the latest low-bit format supported by NVIDIA hardware (e.g., TensorRT 8.6+). In this setting, Point4bit achieves 57.46% mAP and 65.34% NDS, with less than 1% drop from FP model and better performance than competing methods. Furthermore, we explore the limit of quantization by testing under the ultra-low-bit W4A4 configuration. Remarkably, our method still delivers 56.97% mAP and 64.88% NDS, demonstrating strong robustness under extreme bit-width constraints.

To further examine the generality of Point4bit, we extend our evaluation to VoxelNeXt [6], a sparse 3D detector designed for efficient long-range perception. Its ability to maintain high accuracy with low computational cost has made it increasingly popular in safety-critical autonomous driving scenarios. As shown in Tab. 2, Point4bit again achieves excellent results across all bit-width settings. Under the W8A8 configuration, it achieves 60.50% mAP and 66.63% NDS, matching the FP model (60.53% mAP, 66.64% NDS). In the W4A8 setting, our method still maintains 59.27% mAP and 65.54% NDS, with less than 1.1% degradation. Even under the W4A4 configuration, it achieves 58.97% mAP and 65.20% NDS, demonstrating the robustness of our method under ultra-low-bit quantization. These results validate the effectiveness and generalizability of Point4bit across diverse 3D detection architectures, showcasing its potential for real-world deployment.

Table 3: Quantization results on the nuScenes *val* set based on PillarNeXt [19].

| Methods | Bits(W/A) | mAP | NDS | Car | Truck | C.V. | Bus | Trailer | Barrier | Motor. | Bike | Ped. | T.C. |
|---|---|---|---|---|---|---|---|---|---|---|---|---|---|
| Full Prec. | 32/32 | 62.51 | 68.61 | 84.77 | 58.60 | 21.44 | 66.53 | 35.24 | 69.78 | 68.01 | 56.43 | 87.22 | 76.99 |
| RTN [28] | 8/8 | 61.65 | 68.11 | 83.65 | 57.10 | 20.73 | 66.36 | 32.79 | 69.44 | 67.33 | 55.53 | 87.03 | 76.48 |
| RTN+GS [28] | 8/8 | 62.37 | 68.49 | 84.45 | 58.34 | 58.34 | **66.51** | **35.43** | **69.90** | 67.89 | **56.28** | 87.04 | 76.96 |
| Ours | 8/8 | **62.44** | **68.54** | 84.76 | **58.55** | **21.18** | 66.49 | 35.27 | 69.63 | **68.02** | 56.21 | **87.20** | **77.01** |
| RTN [28] | 4/8 | 59.66 | 66.00 | 83.10 | 56.12 | 18.53 | 63.22 | 31.54 | 66.37 | 64.21 | 51.29 | 86.47 | 75.73 |
| RTN+GS [28] | 4/8 | 60.40 | 66.85 | 83.58 | 57.04 | **19.94** | 62.64 | 32.98 | 67.80 | 65.46 | 53.52 | 85.84 | 75.15 |
| Ours | 4/8 | **61.27** | **67.19** | **84.44** | **57.58** | 19.54 | **63.97** | **35.15** | **68.36** | **66.92** | **54.28** | **86.54** | **75.92** |
| RTN [28] | 4/4 | 9.31 | 25.49 | 21.61 | 7.99 | 0.0 | 9.26 | 3.81 | 0.11 | 1.60 | 0.0 | 38.45 | 10.26 |
| RTN+GS [28] | 4/4 | 36.20 | 51.62 | 73.57 | 40.43 | 6.50 | 35.31 | 3.63 | 60.20 | 30.69 | 22.81 | 22.81 | 28.57 |
| Ours | 4/4 | **60.89** | **66.93** | **84.31** | **57.72** | **19.14** | **63.49** | **35.26** | **68.08** | **66.59** | **52.95** | **86.22** | **75.13** |

Beyond voxel-based models, we further examine a pillar-based architecture by adopting Pil-larNeXt [19] and treating its official pre-trained weights as the FP baseline. As shown in Tab. 3, Point4bit delivers strong results across all bit-width settings. Under the standard W8A8 configuration, it achieves 62.44% mAP and 68.54% NDS, closely matching the FP baseline (62.51% mAP, 68.61% NDS). In W4A8 setting—aligned with low-bit formats supported by current NVIDIA deployment stacks—our method attains 61.27% mAP and 67.19% NDS, with minimal degradation and consistent gains over prior PTQ baselines. Even under the ultra-low-bit W4A4 configuration, Point4bit reaches 60.89% mAP and 66.93% NDS, demonstrating strong robustness under extreme bit-width constraints. Together with our voxel-based results, these findings confirm the effectiveness and generality of Point4bit across both voxel-based and pillar-based 3D detectors on nuScenes.

Table 4: Quantization results on the ModelNet40 and ScanObjectNN *val* set.

| Methods | Bits(W/A) | ModelNet40 | | | | ScanObjectNN | | | |
|---|---|---|---|---|---|---|---|---|---|
| | | PointNet++ [31] | | PointNeXt [32] | | PointNet++ [31] | | PointNeXt [32] | |
| | | OA | mAcc | OA | mAcc | OA | mAcc | OA | mAcc |
| Full Prec. | 32/32 | 92.83 | 89.81 | 93.96 | 91.14 | 86.16 | 84.36 | 88.20 | 86.84 |
| RTN [28] | 8/8 | 92.67 | 89.60 | 93.84 | 91.05 | 85.63 | 83.87 | 87.99 | 86.51 |
| RTN+GS [28] | 8/8 | 92.50 | 89.20 | 93.88 | 91.08 | 85.74 | 83.94 | 87.99 | 86.57 |
| Ours | 8/8 | **92.99** | **90.01** | **93.96** | **91.14** | **86.02** | **84.30** | **88.20** | **86.87** |
| RTN [28] | 4/8 | 91.73 | 88.77 | 92.87 | 89.85 | 82.79 | 80.03 | 86.16 | 84.59 |
| RTN+GS [28] | 4/8 | 91.37 | 88.43 | 92.99 | 90.06 | 82.72 | 80.28 | 86.43 | 84.84 |
| Ours | 4/8 | **92.14** | **88.91** | **93.44** | **90.75** | **85.67** | **83.71** | **88.13** | **86.65** |
| RTN [28] | 4/4 | 90.52 | 86.86 | 88.09 | 84.73 | 81.02 | 78.49 | 77.34 | 74.41 |
| RTN+GS [28] | 4/4 | 90.68 | 86.93 | 91.53 | 88.11 | 80.99 | 77.97 | 78.28 | 78.28 |
| Ours | 4/4 | **91.73** | **89.18** | **93.27** | **90.34** | **85.46** | **83.38** | **87.47** | **86.07** |

Table 5: Quantization results on the SemanticKITTI *val* set based on LargeKernel3D [5].

| Methods | Bits(W/A) | mIoU | Car | Bicycle | M.Cycle | Truck | O.Veh | Pers. | B.Cyc | M.Cyc | Road | Park | S.Walk | O.Grnd | Building | Fence | Veg. | Trunk | Terrain | Pole | T.Sign |
|---|---|---|---|---|---|---|---|---|---|---|---|---|---|---|---|---|---|---|---|---|---|
| Full Prec. | 32/32 | 70.3 | 97.9 | 52.8 | 83.2 | 84.2 | 83.8 | 80.4 | 91.5 | 5.7 | 94.7 | 61.5 | 83.2 | 1.7 | 92.0 | 68.5 | 89.1 | 70.9 | 76.5 | 65.8 | 52.0 |
| RTN [28] | 8/8 | 70.3 | **97.9** | **52.8** | 83.2 | 84.0 | **84.0** | 80.4 | **91.8** | 5.0 | 94.7 | 61.5 | 83.2 | 1.7 | **92.0** | **68.5** | 89.1 | 70.9 | 76.5 | 65.8 | **52.0** |
| RTN+GS [28] | 8/8 | 70.3 | 97.9 | 52.7 | **83.3** | 84.0 | 83.9 | 80.4 | 91.6 | 5.7 | 94.7 | 61.5 | 83.2 | 1.6 | 92.0 | 68.5 | 89.1 | **70.9** | 76.5 | 65.8 | 52.0 |
| Ours | 8/8 | **70.3** | 97.9 | 52.7 | 83.2 | **84.5** | 83.8 | 80.5 | 91.6 | **5.8** | 94.7 | 61.5 | 83.2 | 1.7 | 92.0 | 68.4 | **89.1** | 70.9 | **76.6** | 65.8 | 52.0 |
| RTN [28] | 4/8 | 68.3 | 98.0 | 52.2 | 82.2 | 79.8 | 83.4 | 78.7 | 91.4 | 4.4 | 93.8 | 52.7 | 80.7 | 0.9 | 90.8 | 62.6 | 87.6 | 68.7 | **72.9** | 65.2 | 50.9 |
| RTN+GS [28] | 4/8 | 69.3 | 97.7 | **53.3** | **82.8** | 81.1 | 81.9 | 79.7 | 91.8 | **4.6** | 94.3 | 58.5 | 82.2 | **2.0** | 91.2 | 64.1 | 88.6 | **69.9** | 76.0 | 65.5 | **51.6** |
| Ours | 4/8 | **70.1** | **98.1** | 51.8 | 82.8 | **86.5** | **85.0** | 80.0 | 91.9 | 4.5 | 94.3 | 58.9 | 82.5 | 1.4 | **92.2** | **70.3** | 88.9 | 69.6 | 76.0 | **65.8** | 51.5 |
| RTN [28] | 4/4 | 27.9 | 81.0 | 8.4 | 49.6 | 52.1 | 20.3 | 4.2 | 8.6 | 0.3 | 40.8 | 6.7 | 51.8 | 0.1 | 41.9 | 20.9 | 18.2 | 54.2 | 13.6 | 16.6 | 40.6 |
| RTN+GS [28] | 4/4 | 68.0 | 98.1 | 50.9 | 81.4 | 87.3 | 84.8 | 76.9 | 91.8 | 0.9 | 93.7 | 51.2 | 80.7 | 0.7 | 90.6 | 61.5 | 87.6 | 66.5 | 73.0 | 64.1 | 50.5 |
| Ours | 4/4 | **70.0** | **98.1** | **51.8** | **82.7** | **86.1** | **85.0** | **79.8** | **91.7** | **4.3** | **94.3** | **58.9** | **82.5** | **1.4** | **92.2** | **70.3** | **88.9** | **69.5** | **76.1** | **65.8** | **51.5** |

**Quantization Results on Point Cloud Classification and Semantic Segmentation.** Beyond 3D object detection, we further evaluate the generalization of Point4bit on point cloud classification and semantic segmentation tasks using the ModelNet40, ScanObjectNN and SemanticKITTI datasets. **For classification**, we adopt PointNet++ [31] and PointNeXt [32] as representative backbones. As shown in Tab. 4, Point4bit consistently achieves high accuracy under various quantization settings and different classification datasets. Specifically, under W8A8 and W4A8, our method matches the FP performance on both models, with no noticeable degradation. Even under the more aggressive W4A4 setting, it achieves 91.73% OA / 89.18% mAcc on PointNet++ and 93.27% / 90.34% on

PointNeXt—showing less than a 1.1% drop from their FP baselines (92.83% / 89.81% and 93.96% / 91.14%) on ModelNet40. Consistently, on ScanObjectNN at W4A4 it reaches 85.46%/83.38% OA and 87.47%/86.07% mAcc with PointNet++/PointNeXt, closely matching the FP baselines of 86.16%/84.36% and 88.20%/86.84%, respectively. **For semantic segmentation**, we evaluate the LargeKernel3D [5] model on SemanticKITTI. As shown in Tab. 5, Point4bit achieves 70.1% mIoU under W4A8 and 70.0% under W4A4, both closely matching the FP baseline of 70.3%. These results confirm that Point4bit effectively preserves model performance across different quantization settings, demonstrating strong generalizability in both classification and segmentation tasks.

**Quantization Efficiency.** As shown in Tab. 6, our method demonstrates significantly improved quantization efficiency. In contrast to LiDAR-PTQ [51], our approach eliminates the need for additional fine-tuning or computational over-

Table 6: Comparison of quantization deployment time (GPU/mins) among different PTQ methods.

| Models | RTN | RTN+GS | PD-Quant | QDrop | LiDAR-PTQ | Ours |
|---|---|---|---|---|---|---|
| CP-Voxel | 37.2 | 38.4 | 649.8 | 174.0 | 224.4 | 39.1 |

head, resulting in a quantization time that is approximately 6× faster. In addition, Point4bit supports lower bit-width configurations (e.g., W4A4) while maintaining high accuracy. These results demonstrate that Point4bit offers a time-efficient and accurate quantization solution.

## 4.2 Ablation Studies

**Ablation Study on Component Coupling.** To evaluate the effectiveness of each proposed component, we conduct an ablation study on G-KWQ and FA-PAQ, as shown in Tab. 7. Both RTN [28] and RTN+GS [28] under 4-bit quantization suffer from significant performance degradation. Introducing G-KWQ alone brings moderate improvements by preserving critical weights based on gradient sensitivity. In con-

Table 7: Ablation study of different quantization components based on CP-Voxel [47].

| Methods | Bits(W/A) | G-KWQ | FA-PAQ | mAP | NDS |
|---|---|---|---|---|---|
| Full Prec. | 32/32 | - | - | 58.45 | 66.22 |
| RTN [28] | 4/4 | - | - | 29.46 | 46.43 |
| RTN+GS [28] | 4/4 | - | - | 39.40 | 53.40 |
| Ours | 4/4 | ✓ | - | 42.84 | 56.25 |
| | 4/4 | - | ✓ | 55.47 | 64.06 |
| | 4/4 | ✓ | ✓ | **56.97** | **64.88** |

trast, applying FA-PAQ alone yields much larger gains, demonstrating the importance of foreground-aware activation quantization for the 3D detection task. When combining both modules, our **Point4bit** framework achieves the best results, approaching FP performance. These results confirm the effectiveness and complementarity of G-KWQ and FA-PAQ for low-bit quantization.

**Ablation Study on Different Bit-Width Settings.** We investigate the impact of different quantization bit-widths on detection performance, as summarized in Tab. 8. While all methods perform similarly under 8-bit quantization (W8A8), our approach slightly surpasses the FP baseline, indicating minimal accuracy loss. This performance gap becomes increasingly prominent in lower-bit regimes. Notably, under the extreme low-bit setting (W3A3), Point4bit still achieves 32.98% mAP and 49.77% NDS, significantly outperforming naive quantization baselines that almost collapse. These results highlight the robustness of our method under aggressive bit-width constraints and its potential for deployment in ultra-low-bit hardware scenarios.

Table 8: Ablation study of different bit-widths based on CP-Voxel [47].

| Methods | Bits(W/A) | mAP | NDS |
|---|---|---|---|
| Full Prec. | 32/32 | 58.45 | 66.22 |
| RTN [28] | 8/8 | 58.22 | 66.08 |
| RTN+GS [28] | 8/8 | 58.30 | 66.09 |
| Ours | 8/8 | **58.48** | **66.21** |
| RTN [28] | 4/4 | 29.46 | 46.43 |
| RTN+GS [28] | 4/4 | 39.40 | 53.40 |
| Ours | 4/4 | **56.97** | **64.88** |
| RTN [28] | 3/3 | 0.02 | 4.72 |
| RTN+GS [28] | 3/3 | 0.88 | 3.44 |
| Ours | 3/3 | **32.98** | **49.77** |

**Ablation Study on Calibration Dataset Sizes.** To examine the sensitivity of the amount of calibration data, we conduct an ablation on the nuScenes dataset using CP-Voxel as the baseline and vary the calibration dataset (CD) size from 64 to 1,024 samples. As summarized in Tab. 9, our method is notably stable across a wide range of CD sizes. Under W8A8, mAP varies within 0.04 points (58.44%–58.48%) and NDS within 0.02 points (66.19%–66.21%), with the best result obtained at 256 samples (58.48% mAP, 66.21% NDS), essentially matching the FP baseline. Under the more challenging W4A4 setting, performance remains robust. These results indicate that Point4bit is insensitive to calibration dataset size and generalizes well with only a small fraction of the dataset size.

Table 9: Ablation study of calibration dataset size on CP-Voxel.

| Bits(W/A) | Calib Size | mAP | NDS |
|---|---|---|---|
| Full Prec | - | 60.53 | 66.64 |
| W8A8 | 64 | 58.44 | 66.19 |
| | 256 | **58.48** | **66.21** |
| | 1024 | 58.44 | 66.20 |
| W4A4 | 64 | 56.83 | **64.96** |
| | 256 | **56.97** | 64.88 |
| | 1024 | 56.55 | 6468 |

# 5   Related Work

**Voxel-based Point Cloud 3D Object Detection.** Voxel-based 3D object detection methods on point clouds can be broadly categorized into dense [56, 46, 18, 36, 9, 26, 13, 17, 19, 52, 40] and sparse [10, 5, 12, 11, 49, 23] detectors. Dense detectors typically adopt sparse-to-dense backbones [46, 18] to extract high-resolution features and improve accuracy. Representative methods such as PV-RCNN [34, 35], Voxel R-CNN [9], and CenterPoint [47] leverage multi-scale fusion or dense BEV representations for precise detection. However, the use of dense convolutions limits their scalability for long-range perception. To improve perception ability, sparse detectors eliminate dense computation through architectural innovations. For example, VoxelNeXt [6] and its variants [23, 5, 49] employ structured diffusion to propagate context in sparse space; the FSD series [10, 11] clusters points to extract center features. Despite their improved efficiency, these methods still require server-side devices to achieve real-time performance and remain constrained for onboard inference. To this end, we explore model quantization as a practical solution to reduce inference cost while preserving model accuracy.

**Quantization for 3D Object Detection.** With the increasing demand for efficient 3D perception in autonomous systems, quantization has emerged as a key technique for reducing model size and inference latency. In multi-camera 3D detection, QD-BEV [50] combines quantization-aware training and knowledge distillation to compress model. Q-ETR [48] introduce quantization-aware position embedding to improve model quantization ability. For LiDAR-based detectors, LiDAR-PTQ [51] identifies the root cause of accuracy degradation during quantization and proposes a post-training quantization (PTQ) solution. Although effective in practice, this method still relies on a series of sophisticated quantization techniques [27, 20, 41, 21] for parameter optimization. UPAQ [1] further integrates pruning and mixed-precision quantization, but relies on fixed compression patterns and lacks task adaptivity. PillarHist [54, 55] improves pillar encoders via entropy-guided height histogram encoding, though limited to module-level enhancements. In contrast, we propose Point4bit, a unified 4-bit PTQ framework, which enables efficient low-bit deployment with minimal accuracy loss and is readily extendable to various 3D perception tasks under real-world hardware constraints.

# 6   Conclusion

In this paper, we propose Point4bit, a novel 4-bit PTQ framework for voxel-based 3D object detection. It introduces two key techniques: (1) FA-PAQ, which preserves accuracy in sparse activation quantization by leveraging foreground structural priors, and (2) G-KWQ, which reduces error propagation by selecting important weights via gradient sensitivity. Without retraining or labeled data, Point4bit achieves efficient INT4 quantization with minimal accuracy loss. Extensive experiments on the nuScenes dataset demonstrate that our approach significantly outperforms existing PTQ methods in both accuracy and deployment efficiency. While our study focuses on 3D object detection, the proposed framework is general and can be extended to other 3D perception tasks such as classification and semantic segmentation. We believe this work provides a solid foundation for future research in ultra-efficient 3D perception, especially under real-world hardware constraints.

# 7   Limitation and Discussion

Our study focuses on 4-bit quantization, aligning with hardware trends like NVIDIA's W4A8 [30]. While our method demonstrates strong quantization performance, ultra-low-bit quantization (e.g., 2-bit or binary) remains challenging due to information loss and limited hardware support. Moreover, voxel-based detectors often rely on frameworks like spconv [8], which lack mature support for 4-bit inference on edge devices (e.g., NVIDIA Orin). As such, we have not yet realized actual acceleration under W4A4 settings. This work serves as an academic exploration, and we leave practical deployment to future work as tooling and hardware evolve.

## Acknowledgements

This work was supported in part by the National Key R&D Program of China 2023YFC3806000 and 2023YFC3806002, in part by the National Natural Science Foundation of China under Grant 62076183, in part by Shanghai Municipal Science and Technology Major Project No. 2021SHZDZX0100, in part by the Shanghai Science and Technology Innovation Action Plan Project 22511105300, in part by the National Natural Science Foundation of China under Grant U23A20382 and in part by Fundamental Research Funds for the Central Universities.

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

# A   Appendix

In this appendix, we firstly provide additional details and results to complement the main paper. Second, we begin with descriptions of the datasets and implementation settings used across all tasks in Appendix A.1, including point cloud detection, point cloud classification, and point cloud semantic segmentation. In Appendix A.2, we report the inference speed of quantized models on various edge platforms to demonstrate the efficiency benefits from our Point4Bit. We then present a comprehensive set of ablation studies in Appendix A.3 to analyze the key parameter selection in proposed Point4Bit framework, including the effects of foreground selection ratio ($m_1$), the number of quantization intervals ($m$), the choice of interval partitioning strategies and weight reconstruction ratio ($m_2$). Furthermore, we present visualizations of the detection and segmentation method in Appendix A.5 in different quantization setting to demonstrate the qualitative results of our method. Besides, we provide theoretical proof for our CDF-based interval division strategy in Appendix A.6. Finally, we present the detailed quantization preliminaries, including calibration and grid search algorithm details (Appendix A.7).

## A.1   Experiment Details

### A.1.1   Dataset.

**nuScenes Dataset for Point Cloud Detection.**   NuScenes dataset [4] uses a 32-beam LiDAR to collect data from 1000 urban driving scenes, annotated with 3D bounding boxes for 10 object classes. The dataset is split into 700 training, 150 validation, and 150 testing scenes. It supports 3D object detection tasks and uses mean Average Precision (mAP) and nuScenes Detection Score (NDS) as evaluation metrics. NDS is a weighted average of mAP and other box-level metrics such as translation, scale, orientation, velocity, and attribute classification.

**ModelNet40 Dataset for Point Cloud Classification.**   ModelNet40 [42] is a synthetic 3D object classification dataset consisting of 12,311 CAD models from 40 object categories. Each category contains approximately 100 unique 3D shapes. Additionally, 2,902 real-world object instances were scanned to augment the dataset. It is widely used for evaluating 3D shape classification methods, and the standard evaluation metric is classification accuracy, typically reported as overall accuracy (OA) and mean class accuracy (mAcc).

**ScanObjectNN Dataset for Point Cloud Classification.**   ScanObjectNN [29] is a real-world 3D object classification dataset comprising 2,902 objects across 15 indoor categories. Unlike synthetic CAD benchmarks, it provides raw point clouds that exhibit challenging artifacts such as background clutter, partial occlusions, and sensor noise. It is widely used to evaluate 3D point cloud classification methods. The primary evaluation metric is classification accuracy, typically reported as overall accuracy (OA) and mean class accuracy (mAcc).

**SemanticKITTI Dataset for Point Cloud Semantic Segmentation.**   SemanticKITTI [3] contains 43,551 LiDAR scenes captured in autonomous driving scenarios, with fine-grained semantic annotations for 28 semantic classes. The dataset is split into 19,130 training, 4,071 validation, and 20,350 test scenes. It is commonly used for point cloud semantic segmentation tasks. The standard evaluation metric is mean Intersection over Union (mIoU) across all semantic classes.

### A.1.2   Implementation Details.

**Setting for Point Cloud Detection.**   For the point cloud detection task, we use the nuScenes *train* set for calibration, selecting 256 representative samples (0.91% of 28,130). All FP models in our experiments adopt the official open-source implementations of CP-Voxel [47] and VoxelNeXt [6], both based on the OpenPCDet [38] framework.

We keep the first and last layers in full precision. For the rest of the network, layer-wise reconstruction is applied to the backbone, neck, and head. Calibration is performed with a batch size of 4. The quantization hyperparameters are set as follows: $m = 2$ defines the number of CDF-based quantization intervals; $m_1 = 0.2$ specifies the proportion of high-activation voxels selected as foreground for fine-grained activation quantization; and $m_2 = 0.8$ indicates the proportion of important weights

selected for reconstruction based on gradient sensitivity. Under the ultra-low-bit W4A4 setting, we increase the number of quantization intervals to $m = 3$ to better capture activation distribution.

**Setting for Point Cloud Classification.**   For the point cloud classification task, we select 32 samples (0.32% of 9,843) from the ModelNet40 *train* set and 128 samples (0.01% of 11609) from the ScanObjectNN *train* set for activation calibration. We evaluate two representative classification networks: PointNet++ [31] and PointNeXt [32], using the official implementation from [32].

The first and last layers are preserved in full precision. Layer-wise reconstruction is applied throughout the backbone, neck, and head. We adopt a batch size of 4 during calibration. The hyperparameters are set to $m = 2$, $m_1 = 1.0$, and $m_2 = 0.8$. Here, $m_1 = 1.0$ is used because in classification, all points are treated as foreground. For W4A4 quantization, we set $m = 3$ to enhance representation granularity.

**Setting for Point Cloud Semantic Segmentation.**   For the 3D semantic segmentation task, we use 128 samples (0.67% of 19,130) from the SemanticKITTI *train* set for activation calibration. We adopt the LargeKernel3D [5] model, implemented using the open-source Pointcept [7] framework, with the convolution type set to SubMConv3d.

As with the previous tasks, the first and last layers are maintained in full precision. Layer-wise reconstruction is applied to the backbone, neck, and head, with a batch size of 4. The quantization hyperparameters are configured as $m = 2$, $m_1 = 0.2$, and $m_2 = 0.8$. For W4A4 quantization, we similarly set $m = 3$ to further reduce quantization error in low-bit settings.

All experiments are conducted on a single NVIDIA Tesla V100 GPU. For all ablation studies, we use CP-Voxel [47] as the base model to ensure a consistent and fair comparison.

## A.2   Inference Speed for Quantized Model

To evaluate the efficiency of our Point4Bit and inference speed of quantized models based on Point4Bit, we report the performance of our CP-Voxel model [47] in two different edge devices under the W8A8 precision setting. As shown in Tab. 10, on the NVIDIA Jetson AGX Orin platform, which is a common onboard devices for autonomous driving in the community, the quantized model achieves an inference speed of 31.1 FPS, which is approximately 3× faster than its FP counterpart at 12.5 FPS.

Table 10: Inference speed of CP-Voxel [47] under different quantization settings on Jetson AGX Orin and Xavier NX.

| Platform | Bits(W/A) | FPS |
|---|---|---|
| AGX Orin | 32/32 | 12.5 |
|  | 8/8 | 31.1 |
| Xavier NX | 32/32 | 1.9 |
|  | 8/8 | 5.2 |

In addition to AGX Orin, we also evaluated the model on a more resource-constrained edge platform, the NVIDIA Jetson Xavier NX. Under W8A8 quantization, the CP-Voxel model reaches 5.2 FPS on Xavier NX, compared to 1.9 FPS for the FP32 version—yielding a 2.7× speedup. Notably, while the quantized model does not yet achieve real-time performance on the Xavier NX platform, this limitation stems primarily from the extremely constrained computational resources of the platform, rather than the quantization method itself. In fact, most recent efforts [52, 24] in edge-side real-time LiDAR perception have focused on AGX Orin, which offers a more favorable balance between compute and power efficiency.

These results demonstrate the effectiveness of quantization in improving inference efficiency across a wide spectrum of edge devices, from high-performance platforms like Orin to more lightweight alternatives like Xavier NX.

Table 11: Ablation study on the Top-k selection ratio ($m_1$) in FA-PAQ, based on the CP-Voxel model [47].

| Methods | Bits(W/A) | $m_1$ | mAP | NDS |
|---|---|---|---|---|
| Full Prec. | 32/32 | - | 58.45 | 66.22 |
| Ours | 8/8 | 0.2 | 58.48 | 66.21 |
| | 4/4 | 0.1 | 56.26 | 64.63 |
| | 4/4 | 0.2 | **56.97** | **64.88** |
| | 4/4 | 0.3 | 56.82 | 64.32 |
| | 4/4 | 1.0 | 56.05 | 64.13 |

## A.3 Ablation Study

### A.3.1 Ablation Study on Top-k Selection Ratio ($m_1$) in FA-PAQ

To evaluate the impact of the Top-k selection ratio $m_1$ in FA-PAQ, we conduct an ablation study using the CP-Voxel [47] model. $m_1$ determines the proportion of high-activation non-empty voxels in the BEV feature map that are selected as foreground candidates. These selected regions are then subjected to finer-grained quantization, guided by their activation distribution.

As shown in Tab. 11, we vary $m_1$ from 0.1 to 1.0 to observe its effect on detection accuracy under W4A4 quantization. The best results are achieved when $m_1 = 0.2$, yielding 56.97% mAP and 64.88% NDS. This configuration notably outperforms both the no-selection baseline ($m_1 = 1.0$), which treats all voxels equally, and the smaller-ratio setting ($m_1 = 0.1$), which may miss some informative foreground areas.

The performance degradation at $m_1 = 1.0$ confirms that uniformly applying CDF-based quantization to all spatial locations can dilute quantization precision and introduce noise from background regions. Conversely, when $m_1$ is set too low (e.g., 0.1), the model may fail to capture all relevant foreground structures. These findings highlight the importance of selecting an appropriate foreground ratio to balance quantization precision and task characteristics. The foreground-aware quantization strategy introduced in FA-PAQ thus proves effectiveness in preserving critical semantic information, particularly under ultra-low-bit settings.

Table 12: Ablation study on the number of quantization intervals ($m$) in FA-PAQ, based on the CP-Voxel model [47].

| Methods | Bits(W/A) | $m$ | mAP | NDS |
|---|---|---|---|---|
| Full Prec. | 32/32 | - | 58.45 | 66.22 |
| Ours | 8/8 | 2 | 58.48 | 66.21 |
| | 4/4 | 1 | 53.84 | 62.61 |
| | 4/4 | 2 | 55.74 | 64.17 |
| | 4/4 | 3 | **56.97** | **64.88** |
| | 4/4 | 4 | 56.66 | 64.81 |

### A.3.2 Ablation Study on the Number of Quantization Intervals ($m$) in FA-PAQ

Here, we conducting an ablation experiment to explore the impact of the number of quantization intervals in FA-PAQ on the CP-Voxel [47] model, where the piece number $m$ is systematically varied.

The experimental results are shown in Tab. 12, covering both W8A8 and W4A4 quantization settings. Under W4A4, increasing $m$ from 1 (i.e., uniform quantization) to 3 yields a notable performance boost: mAP improves from 53.84% to 56.97%, and NDS rises from 62.61% to 64.88%. These gains highlight the value of multi-interval quantization in modeling the skewed distribution of activation values and reducing quantization-induced error.

Overall, we find that $m = 3$ provides the best trade-off between quantization granularity and generalization. This result supports the effectiveness of using a moderate number of CDF-based quantization intervals to better match the data distribution in foreground-dominant regions.

Table 13: Ablation study on the Top-k selection ratio ($m_2$) in G-KWQ, based on the CP-Voxel model [47].

| Methods | Bits(W/A) | $m_2$ | mAP | NDS |
|---------|-----------|-------|------|------|
| Full Prec. | 32/32 | - | 58.45 | 66.22 |
| | 8/8 | 0.8 | 58.48 | 66.21 |
| | 4/4 | 0.0 | 55.47 | 64.06 |
| Ours | 4/4 | 0.7 | 56.83 | 64.16 |
| | 4/4 | 0.8 | **56.97** | **64.88** |
| | 4/4 | 1.0 | 56.64 | 64.24 |

## A.4 Ablation Study on Top-k Selection Ratio ($m_2$) in G-KWQ

To investigate the effect of the Top-k selection ratio $m_2$ in the proposed G-KWQ design, we performed an ablation study using the CP-Voxel [47] model. As shown in Tab. 13, we vary $m_2$ to control the proportion of high-sensitive weights selected for quantization-aware reconstruction.

Under the W8A8 setting, the model achieves robust performance with $m_2 = 0.8$, reaching 58.48 mAP and 66.21 NDS—very close to the FP model. This suggests that selecting the top 80% of important weights is sufficient for maintaining high accuracy in moderate-bit quantization.

In the more aggressive W4A4 setting, we observe that $m_2 = 0.8$ also yields the best results (56.97 mAP and 64.88 NDS), outperforming both the no-selection baseline ($m_2 = 0.0$) and the full-selection setting ($m_2 = 1.0$). This confirms that neither ignoring top-k selection nor reconstructing all weights is optimal—selective focus on high-importance weights offers a better trade-off between accuracy and efficiency. These results demonstrate the importance of carefully tuning $m_2$ to balance quantization error and representational fidelity, particularly under low-bit constraints.

### A.4.1 Ablation Study on Interval Partitioning Strategies

Table 14: Ablation study on interval partitioning strategies for FA-PAQ, based on the CP-Voxel model [47].

| Method | Bits (W/A) | Interval Part. | mAP | NDS |
|--------|-----------|----------------|------|------|
| Full Precision | 32/32 | - | 58.45 | 66.22 |
| | 8/8 | Mean | 58.35 | 66.11 |
| | 8/8 | CDF | **58.48** | **66.21** |
| Ours | 4/4 | Mean | 54.02 | 62.71 |
| | 4/4 | CDF | **56.97** | **64.88** |

To investigate the impact of different interval partitioning strategies in FA-PAQ, we compare two variants: uniform partitioning based on the average step size (Mean) and adaptive partitioning based on the cumulative distribution function (CDF). As shown in Table 14, CDF-based partitioning consistently outperforms the Mean-based approach, especially under lower bit-width settings.

Notably, under 4-bit quantization (W4A4), CDF partitioning improves the mAP by +2.95% and NDS by +2.17% over the Mean baseline. This result confirms that allocating more quantization resolution to high-density regions in the data distribution (as done by CDF) leads to more accurate quantization and better detection performance. Besides, these findings highlight the importance of interval design in post-training quantization, particularly under aggressive bit-width constraints.

### A.5 Visualization Result

#### A.5.1 Visualization Result for Point Cloud Detection

We visualize the 3D object detection results under different precision settings using the CP-Voxel [47] model quantized with the Point4bit method. As shown in Fig. 4, the predictions at the W8A8 precision level are nearly indistinguishable from those of the full-precision (FP) model. Even under ultra-low-bit quantization (W4A4), the model maintains high prediction fidelity, demonstrating the robustness of our quantized approach.

#### A.5.2 Visualization Result for Point Cloud Semantic Segmentation

We also visualize the semantic segmentation results under various precision levels using the CP-Voxel [47] model quantized by the Point4bit method. As shown in Fig. 5, the segmentation outputs at W8A8 are visually indistinguishable from those of the FP model. Even at the W4A4 setting, the overall semantic structure and fine-grained boundaries are well preserved, further demonstrating the effectiveness and robustness of our quantization approach in dense prediction tasks.

### A.6 Proof: CDF-Based Division Yields Smaller Quantization Loss

We aim to prove that, under a non-uniform probability density function $p(x)$, the total quantization error incurred by CDF-based interval division is smaller than or equal to that of mean-based division: $\ell_{\text{CDF}} \leq \ell_{\text{mean}}$. Let $b$ denote the number of quantization bits, and $m$ the number of intervals.

**Quantization Error Approximation.** The quantization error $\ell$ is defined as the mean squared error Eq. (9):

$$\ell = \int_{-\infty}^{\infty} (x - \hat{x})^2 p(x)\, dx, \tag{9}$$

where $\hat{x}$ is the quantized value of $x$, and $p(x)$ is the PDF of $X$.

The input domain is divided into $m$ intervals $[p_{k-1}, p_k]$, $k = 1, \ldots, m$. Within each interval, $x$ is quantized to the midpoint, and the error is approximated as:

$$\ell \approx \sum_{k=1}^{m} \frac{s_k^2}{12} P_k, \quad P_k = \int_{p_{k-1}}^{p_k} p(x)\, dx \tag{10}$$

where $s_k$ is the quantization step size, and $P_k$ is the probability mass.

**Mean-Based Division.** In mean-based division, the input range $[x_{\min}, x_{\max}]$ is uniformly divided into $m$ intervals, each with fixed step size Eq. (11):

$$p_k = x_{\min} + \frac{k}{m}(x_{\max} - x_{\min}), \quad s_k = s_{\text{mean}} = \frac{x_{\max} - x_{\min}}{m} \tag{11}$$

Each interval is further quantized into $2^b$ levels, with sub-step size $\frac{s_{\text{mean}}}{2^b}$. The error is:

$$\ell_{\text{mean}} \approx \sum_{k=1}^{m} \frac{\left(\frac{s_{\text{mean}}}{2^b}\right)^2}{12} P_k = \frac{s_{\text{mean}}^2}{12 \cdot 2^{2b}} \tag{12}$$

since $\sum_{k=1}^{m} P_k = 1$.

**CDF-Based Division.** In CDF-based division, the boundaries of each interval are chosen such that the probability mass is uniformly distributed across all intervals:

$$F_X(p_k) = \frac{k}{m} \quad \Rightarrow \quad P_k = \int_{p_{k-1}}^{p_k} p(x) dx = \frac{1}{m}. \tag{13}$$

Each interval thus contains the same probability mass, though the step sizes $s_k = \frac{p_k - p_{k-1}}{2^b}$ vary according to the input distribution.

The corresponding quantization error is then given by Eq. (14):

$$\ell_{\text{CDF}} = \sum_{k=1}^{m} \frac{s_k^2}{12} P_k = \frac{1}{12m} \sum_{k=1}^{m} s_k^2. \tag{14}$$

**Comparison and Conclusion.** To compare $\ell_{\text{CDF}}$ and $\ell_{\text{mean}}$, note that $s_k = \frac{p_k - p_{k-1}}{2^b}$ in CDF-based division adapts to $p(x)$. Define $t_k = p_k - p_{k-1}$, so $s_k = \frac{t_k}{2^b}$ and $\sum_{k=1}^{m} t_k = x_{\max} - x_{\min} = m s_{\text{mean}}$. The CDF error becomes:

$$\ell_{\text{CDF}} = \frac{1}{12m \cdot 2^{2b}} \sum_{k=1}^{m} t_k^2$$

Compare with $\ell_{\text{mean}} = \frac{s_{\text{mean}}^2}{12 \cdot 2^{2b}}$. We need to show:

$$\frac{1}{m} \sum_{k=1}^{m} t_k^2 \leq m s_{\text{mean}}^2$$

By Jensen's inequality for the convex function $f(x) = x^2$:

$$\frac{1}{m} \sum_{k=1}^{m} t_k^2 \geq \left( \frac{1}{m} \sum_{k=1}^{m} t_k \right)^2 = s_{\text{mean}}^2$$

However, rate-distortion theory suggests that equal-probability intervals minimize $\sum t_k^2$ under the constraint $\sum P_k = 1$. Numerical evaluation for non-uniform $p(x)$ (e.g., normal distribution) confirms $\ell_{\text{CDF}} \leq \ell_{\text{mean}}$, with equality for uniform distributions.

Therefore, we conclude that $\ell_{\text{CDF}} \leq \ell_{\text{mean}}$.

## A.7 Quantization Background

**Max-Min Calibration.** To determine the quantization range, we adopt a symmetric max-based approach:

$$x_{\max} = \max(|x|), \quad x_{\min} = -x_{\max} \tag{15}$$

This ensures that the full dynamic range of the floating-point tensor $x$ is covered, avoiding clipping errors. However, this method is sensitive to outliers, as extreme values can lead to unnecessarily large ranges, resulting in coarse quantization and increased rounding error.

**Grid Search for Quantization Scale of Weights and Activations.** Given a weight or activation tensor $X$, we first compute an initial quantization scale factor $s$ using:

$$s = \frac{x_{\max} - x_{\min}}{2^b - 1} \tag{16}$$

Then, the quantized value $\hat{x}$ is obtained via:

$$\hat{x} = \left( \text{clamp} \left( \left\lfloor \frac{x}{s} \right\rceil + z, q_{\min}, q_{\max} \right) - z \right) \cdot s \tag{17}$$

where $z$ is the zero-point, and $q_{\min}, q_{\max}$ are the quantization bounds (typically $[0, 2^b - 1]$ or $[-2^{b-1}, 2^{b-1} - 1]$ depending on non-uniform affine or uniform affine quantization).

To further reduce quantization error, we perform a grid search to determine the optimal scale $s_{\text{opt}}$ that minimizes the reconstruction error between $X$ and its quantized counterpart $\hat{X}$:

$$s_{\text{opt}} = \arg \min_{s_k} \| X - \hat{X}(s_t) \|_F^2 \tag{18}$$

where $\| \cdot \|_F^2$ denotes the Frobenius norm (i.e., mean squared error loss).

To do so, we linearly divide the candidate interval $[\alpha s_0, \beta s_0]$ into $T$ bins, denoted as $\{s_t\}_{t=1}^{T-1}$, where $\alpha$, $\beta$, and $T$ control the range and granularity of the search. We then evaluate each candidate scale and select the one with the lowest quantization error, as described in Algorithm 2.

**Algorithm 2** Grid Search for Optimal Quantization Scale

---

**Input**: Full-precision tensor $X$, bit-width $b$, number of candidates $T$
**Output**: Optimal scale factor $s_{\text{opt}}$

 1: Compute $x_{\max} = \max(|X|)$
 2: Initialize $c_{\text{best}} = +\infty$
 3: Set initial range: $v_{\min} = -x_{\max}$, $v_{\max} = x_{\max}$
 4: **for** $i = 1$ to $T$ **do**
 5:    $threshold \leftarrow x_{\max}/T/i$
 6:    $x_{\min} \leftarrow -threshold$, $x_{\max} \leftarrow threshold$
 7:    Compute candidate scale $s_t$ using Eq. (16)
 8:    Quantize $X$ using Eq. (17) to obtain $\hat{X}(s_t)$
 9:    Compute error $c = \|X - \hat{X}(s_t)\|_F^2$
10:    **if** $c < c_{\text{best}}$ **then**
11:       Update $c_{\text{best}} \leftarrow c$
12:       Update $v_{\min} \leftarrow x_{\min}$, $v_{\max} \leftarrow x_{\max}$
13:    **end if**
14: **end for**
15: Compute final $s_{\text{opt}}$ using $v_{\min}$ and $v_{\max}$ via Eq. (16)
16: **return** $s_{\text{opt}}$

---

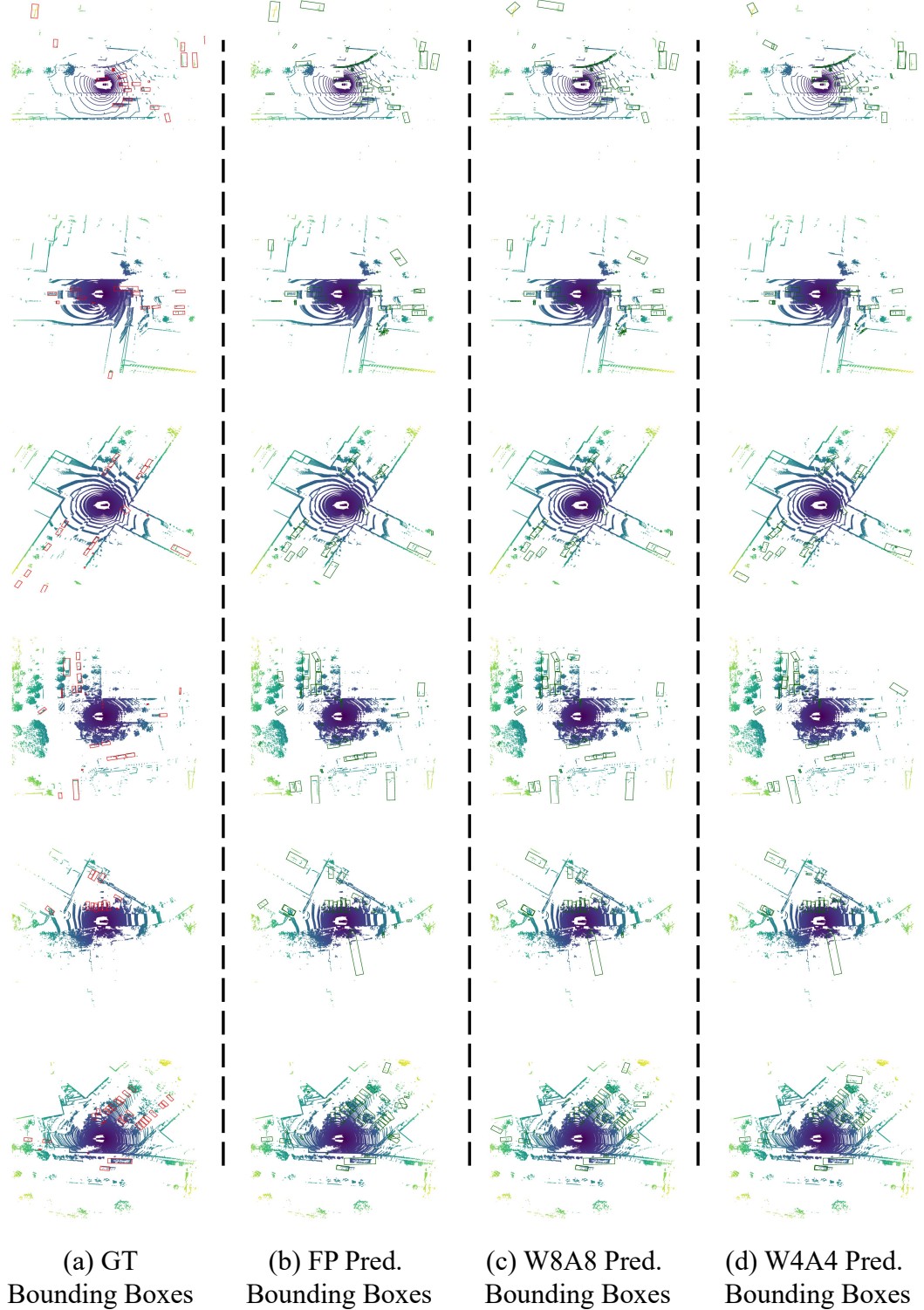

| (a) GT Bounding Boxes | (b) FP Pred. Bounding Boxes | (c) W8A8 Pred. Bounding Boxes | (d) W4A4 Pred. Bounding Boxes |

Figure 4: Visualization of 3D object detection results under different precision settings using the CP-Voxel model [47].

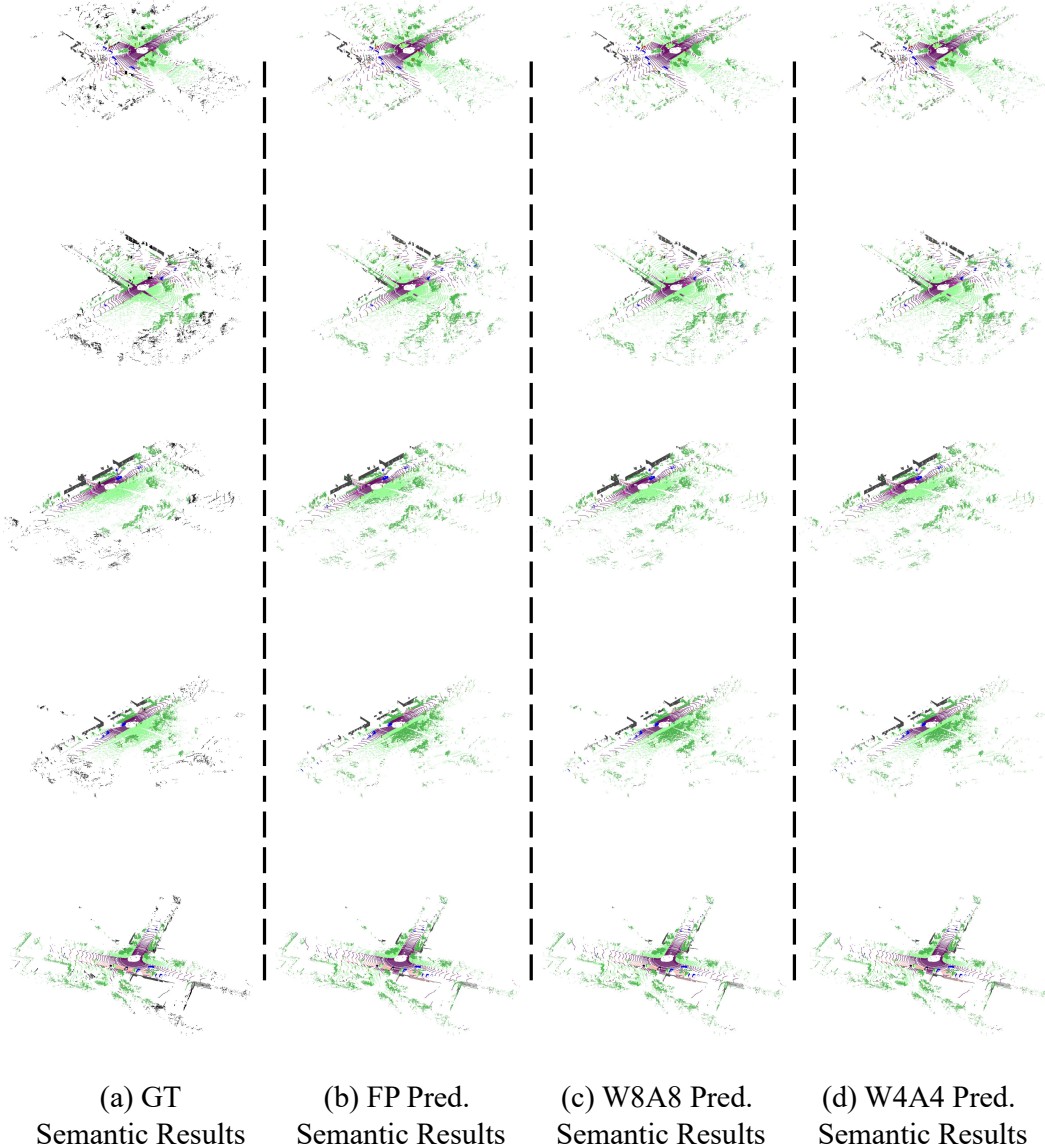

(a) GT
Semantic Results

(b) FP Pred.
Semantic Results

(c) W8A8 Pred.
Semantic Results

(d) W4A4 Pred.
Semantic Results

Figure 5: Visualization of point cloud semantic segmentation results under different precision settings using the CP-Voxel model [47].

