# OpenReview forum: "Point4Bit: Post Training 4-bit Quantization for Point Cloud 3D Detection"
_NeurIPS.cc/2025/Conference — NeurIPS 2025 poster_

### Official Review · Reviewer_Uao9 · 2025-06-27

**Clarity:** 3
**Significance:** 2
**Originality:** 3
**Rating:** 5
**Confidence:** 4

**Summary:**

This paper presents a low-bit quantization method for voxel-based object detection, segmentation and classification in point clouds. It proposes a method that leverages structural information for foreground region aware piecewise activation quantization. It also exploits gradients to quantize weights adaptively, prioritizing task-critical weights. Finer level of quantization is used for weights that are important to the target task. Evaluation is performed across multiple tasks: detection (using mean precision and nuScenes Detection Score), classification (using overall accuracy and mean class accuracy), and segmentation (using mean Intersection-over-Union). Experiments on quantitative comparison are conducted to demonstrate the performance.

**Questions:**

Please see the weaknesses above and consider responding in your rebuttal.

**Ethical Concerns:**

["NO or VERY MINOR ethics concerns only"]

**Final Justification:**

I was already positive about this paper. The authors have addressed all my comments. Even if I do not fully agree regarding the ML insights, I still think this is a good paper and will be of value to the community. Hence, I have increased my score.

**Limitations:**

Yes

**Quality:**

3

**Strengths And Weaknesses:**

Strengths
1.	This paper handles higher quantization level i.e. INT8, than existing methods.

2.	The proposed method is foreground aware and performs adaptive weight quantization.

3.	The piece-wise activation quantization is quite interesting.

4.	The paper is well written and equations are well presented in detail (particularly the gradient-based weight sensitivity analysis)l.

5.	Experiments are performed on multiple tasks (detection, classification, and segmentation) and SOTA results are reported.

Weaknesses
1.	My biggest concern is that the proposed method is quite adhoc and empirical, giving little insights for a machine learning conference such as NeurIPS.

2.	The argument that existing methods only perform INT8 quantization is not valid as it is just the quantization level (hyper-parameter) they chose and not a hard constraint.

3.	Similarly, the argument that previous methods for PTQ require complex training steps that take several hours is also weak. Complexity itself is not a weakness and PTQ is an offline process so several hours is not that much of a delay.

4.	FA-PAQ: Of course, foreground regions will have larger activations but these regions are data (environment) specific. I suspect that this task specific/adaptive quantization may reduce the generalization ability of the network. Can you show results on nuScenes when calibration data comes from SemanticKITTI and vice versa?

5.	How does the method perform at higher bit-width quantization (e.g., 6–8 bits)? Is there a trade-off between precision and runtime savings?

6.	Could the voxelization process itself affect quantization and hence detection performance?

---

> ### Author Rebuttal · Authors · 2025-07-31
>
> We sincerely appreciate your constructive feedback. Below, we provide point-by-point responses, with all revisions incorporated accordingly to improve the quality and clarity of our manuscript.
>
> > **Q1: Insights for a machine learning conference.**
>
> **A1:**   We would like to clarify the core insights of our work, which are significant beyond just empirical performance.
>
> Previous LiDAR-PTQ (ICLR 2024) has demonstrated that exploring the inherent sparsity of point clouds and maintaining the long-distance geometries can enable effective 8-bit post-training quantization for detection models.
>
> Unlike LiDAR-PTQ, we investigate the distributional differences between foreground and background features to push quantization bounds to 4 bits. We posit a more fine-grained principle crucial for ultra-low-bit precision: **Task-critical information in 3D perception models is not uniformly distributed but is highly concentrated in a sparse subset of features corresponding to foreground objects, which is crucial for low-bit quantization.**
>
> Our Point4Bit, is a principled and unified application of this insight, not an ad-hoc collection of techniques:
>
> 1. **For Activations (FA-PAQ):** Our Foreground-aware Piecewise Activation Quantization embodies this principle by systematically identifying and preserving information-rich foreground activations while allowing the less critical background activations to be quantized more aggressively. This goes beyond generic sparsity by leveraging semantic structure.
>
> 2. **For Weights (G-KWQ):** Our Gradient-guided Key Weight Quantization applies this principle to the weight space. It uses gradients to identify weights most sensitive to the task-objective—those responsible for processing critical foreground information—allowing us to preserve them, a necessity for maintaining performance at 4-bit precision.
>
> This principled, importance-aware approach is precisely what enables us to break the 8-bit barrier. While LiDAR-PTQ demonstrated success at 8-bit, its methods lack the semantically-aware prioritization needed for the challenging 4-bit regime. Our work shows that by exploiting the foreground/background feature disparity, it is possible to achieve robust 4-bit PTQ that generalizes across diverse tasks.
>
> We believe this insight—that ultra-low-bit 3D quantization hinges on discerning foreground from background—and the framework to exploit it, constitutes a novel and valuable contribution for the machine learning community.
>
>
> > **Q2: INT8 quantization is not valid as it is just the quantization level.**
>
> **A2:** Thank you for your correction. We agree that quantization bit-width is a hyper-parameter and not a fundamental constraint of prior methods. Our intention was not to claim that these approaches are technically incapable of running at 4-bit, but rather to highlight a practical limitation: their performance collapses when quantizing to 4-bit.
>
> We would like to clarify our main claim: **while prior PTQ methods for 3D point cloud detection achieve strong results at 8-bit, their underlying algorithms are not robust enough for the ultra-low-bit regime.** Simply changing the quantization level to 4-bit results in substantial performance degradation. This is because their quantization schemes treat all parameters and activations with uniform importance—a strategy viable at 8-bit but ineffective at 4-bit, where quantization error is much more pronounced and sensitive features are irreversibly damaged.
>
> In contrast, our method's core contribution is a new, importance-aware quantization strategy (e.g., FA-PAQ and G-KWQ) designed specifically to handle the extreme information loss of 4-bit quantization. By adaptively preserving task-critical features, our approach maintains high performance where others fail.
>
> We have revised the relevant statements in our paper to remove any ambiguity.
>
> >**Q3: Complexity and several hours of offline quantization are not significant weaknesses for PTQ methods.**
>
> **A3:** Thanks for your comments. We agree that for a one-time, offline process, several hours of quantization time can be acceptable.
>
> Our intention was not to overstate the importance of speed alone, but that the "complexity" of prior methods extends beyond computation time to include significant implementation and pipeline overhead. For example, the state-of-the-art method, LiDAR-PTQ, requires a multi-stage layer-wise calibration pipeline. This includes additional steps like generating pseudo-labels from the FP model to supervise quantization parameters. This not only increases the end-to-end quantization time (e.g., 224 minutes vs. our 39 minutes on a V100 GPU) but also makes the process more cumbersome to implement.
>
> While the complexity is not an inherent weakness, our method's simplicity and efficiency offer valuable practical advantages in scenarios like:
> 1. **Frequent Model Updates**: Reducing the turnaround time in production environments where models are frequently updated.
>
> 2. **On-Device or Resource-Constrained Compilation**: Facilitating deployment across diverse edge devices where complex calibration pipelines may be infeasible.
>
> We have revised the content to better frame our contribution around these practical benefits. Thanks again for your helpful feedback.
>
> > **Q4: Provide cross-dataset calibration results to evaluate generalization**
>
> **A4:** Thank you for your question. We would like to clarify the standard and established methodology for Post-Training Quantization (PTQ) that is widely adopted by the research community. The fundamental purpose of the calibration set is to capture a representative statistical distribution of the activations that the model will encounter in its target domain. Therefore, **it is a standard protocol to use a small calibration dataset drawn from the same distribution** as the model's training or validation data. This principle is foundational to nearly all existing PTQ studies, including key works like the Quantization White Paper [23], LiDAR-PTQ [38], and PD-Quant [17], Q-Drop [31].
>
> The proposed experiment—calibrating a nuScenes-trained model with SemanticKITTI data—would introduce a severe domain mismatch. Such a setup produces sub-optimal or misleading quantization results because the calibration process is optimized for an irrelevant data distribution. Consequently, this design does is not a fair or meaningful evaluation of the quantization method's performance on its target task and deviates significantly from established practices.
>
> By following the standard protocol of using in-distribution calibration data, our quantization performance and robustness to calibration set size already provide compelling evidence of our method's generalization ability within its intended domain. We hope this clarifies our methodological and the established context for PTQ evaluation.
>
> > **Q5: Method perform at higher bit-width quantization (e.g., 6–8 bits)? Is there a trade-off between precision and runtime savings?**
>
> **A5:** We provide comprehensive results for 8-bit and 6-bit quantization in Tab.1–4 of our main paper and Tab.1 in the ablation below. Our method achieves nearly lossless performance at these higher bit-widths. For example, on CP-Voxel, our 8-bit model achieves 58.48 mAP and 66.21 NDS, which is virtually identical to the full-precision baseline (58.45 mAP, 66.22 NDS). Even at 6 bits, performance remains high at 58.29 mAP and 65.92 NDS.
>
> Regarding the trade-off: While it is fundamental that lower bit-widths enable greater efficiency at the risk of accuracy, our method provides a **far more favorable trade-off curve** than existing approaches. Generally, increasing bit-width improves accuracy but reduces latency gains, while lower bit-width provides greater speedup at the cost of potential accuracy loss. Our method significantly narrows this trade-off: the near-lossless performance at 8 and 6 bits demonstrates the fundamental robustness of our quantization strategy, which is what enables Point4Bit to successfully push into the challenging 4-bit regime without severe degradation.
>
> **In summary:** Our approach is not a narrow "4-bit solution," but a flexible and robust framework that excels across a range of bit-widths, empowering users to choose the optimal balance between accuracy and efficiency for their deployment needs.
>
> **Tab.1. Ablation Study: Performance at Different Bit-Widths (CP-Voxel).**
>
> | Methods| Bits(W/A) | mAP | NDS |
> |-|-|-|-|
> | Full Prec. | 32/32 | 58.45 | 66.22 |
> | RTN| 6/6  | 54.48 | 63.43 |
> | RTN+GS| 6/6| 57.94 | 65.72 |
> | **Ours** | 6/6| **58.29** | **65.92** |
>
> > **Q6: Could the voxelization process itself affect quantization and hence performance?**
>
> **A6:** Voxelization is a pre-processing step that discretizes raw point clouds and can influence the accuracy of FP detection models (e.g., SECOND[33], VoxelNet[41]).
>
> Our quantization approach is independent of voxelization. As a post-training method, it is applied directly to the trained models and is agnostic to whether the initial features were derived from voxels, pillars, or raw points.
>
> To explicitly validate this independence and demonstrate the general applicability of our method, our evaluation intentionally includes a diverse set of both voxel-based and non-voxelized architectures. Refer to R2Q3, our method is also applicable for Pillar-based detector. Besides, as shown in Tab. 3 and 4 of our main paper, we achieve nearly lossless performance on non-voxelized models for both classification (PointNet++, PointNeXt) and segmentation (LargeKernel3D). For instance, the accuracy drop at W4A4 precision is less than 0.8\%, confirming that the effectiveness of our method is not dependent on the presence of a voxelization pre-processing step.
>
> This demonstrates that our proposed method is a general-purpose quantization solution for 3D point cloud models, not limited to those that rely on a voxel-based representation or are influenced by voxelization operations.

---

> > ### Comment · Reviewer_Uao9 · 2025-08-05
> > **Comments on rebuttal**
> >
> > Thank you for your detailed rebuttal. Most of my concerns have been addressed, especially to Q2, Q3, Q5 and Q6. Please ensure that the final paper is updated accordingly.
> >
> > I still think that the insights mentioned in your response, while significant and practical, are not related to machine learning. It is obvious that "Task-critical information in 3D perception models is not uniformly distributed but is highly concentrated in a sparse subset of features corresponding to foreground objects". However, I will still give you credit for exploiting this and pushing the quantization limit.
> >
> > Regarding Q4, even if the current established and widely adopted methodology for PTQ is to train and test on the same dataset, this does not necessarily present best practice. PTQ techniques should be able to generalize across domains. This common limitation that current PTQ techniques are sensitive to domain shifts should be explicitly highlighted in the paper.

---

> > > ### Author Response · Authors · 2025-08-05
> > >
> > > Thank you for your recognition of our insights and novelty. We're glad our previous responses have addressed most of your concerns.
> > >
> > > **Regarding Q1** (Relevance to Machine Learning), we appreciate your feedback and understand your concern that our insights, though significant and practical, may not appear directly related to machine learning. However, we believe that systematically formalizing and integrating domain-specific observations—such as the concentration of task-critical information in foreground regions—into quantization frameworks constitutes a meaningful machine learning contribution. Many advances in ML arise from transforming intuitive or empirical observations into robust, scalable algorithms. Our work bridges this gap and demonstrates practical benefits for low-bit quantization in 3D model deployment.
> > > **Regarding Q4** (domain sensitivity),We  agree that domain sensitivity is a common limitation for current PTQ methods, including ours. As you suggested, we will clarify this limitation in our final paper. We recognize that improving cross-domain robustness is a valuable direction for the development of Quantization and appreciate you highlighting this point. We will explore it in future work.
> > >
> > > Thank you again for your thoughtful feedback.

---

> > > > ### Comment · Reviewer_Uao9 · 2025-08-05
> > > > **Final comment**
> > > >
> > > > Thank you for your response. All in all, I think this is a good paper and would be valuable for the community. I will increase my final rating to accept.

---

> > > > > ### Author Response · Authors · 2025-08-05
> > > > >
> > > > > Thank you for your positive feedback and for indicating that you would raise your score. We're glad that our responses have addressed your concerns. All the revisions will be incorporated into the final version. We will continue working to further improve Point4bit and our future projects, and look forward to contributing even more to the community.

---

### Official Review · Reviewer_hG7x · 2025-07-01

**Clarity:** 3
**Significance:** 3
**Originality:** 3
**Rating:** 5
**Confidence:** 4

**Summary:**

The paper presents "Point4Bit," a framework designed for post-training 4-bit quantization in point cloud 3D detection. It introduces two key components: a foreground-aware piecewise activation quantization strategy and a gradient-guided key weight quantization method. The proposed approach aims to enhance the performance of voxel-based 3D object detection under low-bit quantization. Through empirical evaluation, the authors demonstrate that Point4Bit effectively minimizes accuracy degradation under ultra-low-bit settings and exhibits strong adaptability for various tasks, including classification and segmentation, making it suitable for resource-constrained environments.

**Questions:**

Please see the weaknesses.

**Ethical Concerns:**

["NO or VERY MINOR ethics concerns only"]

**Final Justification:**

This paper introduces a 4-bit PTQ method that achieves a good balance between accuracy and efficiency, surpassing existing methods. All in all, this paper is technically solid, well written, and practical. Therefore, I give a final rating of accept.

**Quality:**

3

**Strengths And Weaknesses:**

Strengths:

1. This paper proposes a Foreground-aware Piecewise Activation Quantization (FA-PAQ) method, which leverages the structural information of points to alleviate performance degradation caused by activation quantization.

2. A Gradient-guided Key Weight Quantization (G-KWQ) strategy is introduced to preserve task-critical weights that are essential for detection performance.

3. The proposed quantization framework is evaluated across various 3D object detectors, demonstrating its effectiveness and robustness.

4. Beyond 3D object detection, additional tasks such as point cloud classification and semantic segmentation are included to verify the cross-task generalizability of the approach.

5. The paper is clearly written and well-organized, making it easy to follow.

Weaknesses:

1. Although the proposed Point4bit method outperforms previous approaches under 8-bit quantization, comparisons with recent state-of-the-art 4-bit quantization methods would further highlight its advantages.

2. Figure 2 lacks intuitiveness; additional explanation or annotations may help readers better grasp its content.

3. Some variables are introduced without definition or explanation. For instance, the symbol $N$ in Eq. (6) and $W_{ij}^{\ell}$ in Eq. (7) should be clearly defined upon first appearance.

4. The source code is not available online or attached in the supplementary material, which limits the reproducibility of the method.

---

> ### Author Rebuttal · Authors · 2025-07-31
>
> We sincerely appreciate your constructive feedback. Below, we provide point-by-point responses, with all revisions incorporated accordingly to improve the quality and clarity of our manuscript.
>
> >**Q1: Comparison with recent state-of-the-art 4-bit quantization methods.**
>
> **A1:**  Thank you for this important suggestion. We agree that a rigorous comparison with state-of-the-art (SOTA) 4-bit methods is crucial for highlighting the advantages of our approach. We would like to clarify that a primary motivation for our work is that robust 4-bit Post-Training Quantization (PTQ) for 3D point cloud models is a largely unexplored and challenging area. Most existing PTQ methods for this domain (e.g., RTN [23], LiDAR-PTQ [38]) only report results for 8-bit quantization, lacking 4 bits evaluation. This is because the high information density and sensitivity of point cloud features often lead to a drastic collapse in model performance when using conventional techniques at sub-8-bit precision.
>
> Our paper's core contribution is precisely to overcome this barrier. To the best of our knowledge, **Point4Bit establishes one of the first strong and stable baselines for 4-bit PTQ in this domain.** Therefore, due to the scarcity of published 4-bit results, we created the strongest possible baselines for a fair comparison. We compare with available baselines under 4-bit settings in our paper. For 8-bit quantization, we provide comprehensive comparisons with recent state-of-the-art methods in the main paper (Tab.1 and Tab.2), where our approach consistently achieves superior or comparable performance. We believe that by making 4-bit PTQ effective for 3D point cloud models, Point4Bit can set a new benchmark for future research. We look forward to comparing our work with other dedicated 4-bit methods as they are developed by the community.
>
> >**Q2: Figure 2 needs clearer explanation or annotations for better understanding.**
>
> **A2:**  Thank you for your valuable suggestion. To improve the clarity of Fig.2, we have revised its caption and expanded the corresponding explanations in the main paper.
>
> Specifically, we have enhanced the caption of Fig.2 to provide a more comprehensive description:
> 1. **Fig.2 (a) Original Activation Distribution**: Shows the overall distribution of activation values in the feature map of the entire point cloud;
> 2. **Fig.2 (b) Non-Empty Voxel Activation Distribution**: Focuses on non-empty voxels, where the activation values of foreground points are more likely to be higher;
> 3. **Fig.2 (c) Top-K in Non-Empty Voxel Activation Distribution**: The value range is divided into intervals of equal cumulative probability, which serves as the foundation for our adaptive quantization intervals.
>
> These revisions will be updated in the final version. Thank you again for your helpful feedback.
>
> >**Q3: Variables defination.**
>
> **A3:**  Thank you for your careful reading and valuable suggestion.
>
> In Eq. (6), the symbol $N$ has been corrected to $b$, representing the quantization bit-width, consistent with the definition in Eq. (1).
>
> In Eq. (7), we have revised $\alpha_{j}^{\ell}$ to $\alpha_{i}^{\ell}$ to accurately indicate the aggregated task sensitivity of the $i$-th channel in the $\ell$-th layer. We have also explicitly defined $W_{i, j}^{\ell}$ as the $j$-th element of the $i$-th channel in the weight tensor of the $\ell$-th layer.
>
> Furthermore, we have carefully reviewed all mathematical symbols throughout the paper to ensure each is clearly defined upon first use. We believe these revisions will further enhance the clarity and readability of our manuscript.
>
> >**Q4: Source code.**
>
> **A4:**  Thank you for highlighting the importance of reproducibility. We are fully committed to open science and making our work accessible. The code will be made fully public on GitHub upon acceptance.

---

> > ### Comment · Reviewer_hG7x · 2025-08-03
> >
> > Thank you for your responses. The proposed 4-bit quantization method makes sense for the real-world deployment of 3D detectors.

---

> > > ### Author Response · Authors · 2025-08-04
> > >
> > > Thank you sincerely for your encouraging comments and for recognizing the value of our method. Your feedback is highly appreciated and motivates us to further enhance Point4bit as well as our subsequent projects.

---

### Official Review · Reviewer_NnTp · 2025-07-02

**Clarity:** 4
**Significance:** 3
**Originality:** 3
**Rating:** 4
**Confidence:** 3

**Summary:**

This paper proposes Point4Bit, a 4-bit post-training quantization pipeline for voxel-based 3D object detectors, which is claimed as the first 4-bit quantization framework for 3D detection. Two techiniques are introduced to mitigate the performance degradation from quantization: FA-PAQ (Foreground-aware Piecewise Activation Quantization) to focus on computation of foreground and G-KWQ (Gradient-guided Key Weight Quantization) to avoid quantization error back-propagation. Experiments are conducted in nuscenes, modelnet, semantickitti.

**Questions:**

See weakness part.

**Ethical Concerns:**

["NO or VERY MINOR ethics concerns only"]

**Final Justification:**

Overall, the authors have effectively addressed my concerns regarding the effectiveness of the proposed quantization techniques, particularly under challenging conditions (sparse point clouds >20 meters) and the applicability on the stronger baseline of PillarNeXt.

While the method is relatively simple and engineering-oriented, it presents a practical and scalable solution for 4-bit post-training quantization. Given these clarifications and additions, I maintain a borderline accept rating.

**Limitations:**

Yes.

**Quality:**

3

**Strengths And Weaknesses:**

Pros:

1. The experiments are comprehensive across several tasks. The quantization shows >1.5 mAP drop under INT4 quantization compared to prior works like RTN on CenterPoint and VoxelNeXt. The method also demonstrates generalization to LargeKernel3D on SemanticKITTI segmentation tasks.

2. This quantization method is a practical way to deployment, as it requires no labeled data or model training and achieves a 6x speedup in quantization time compared to LIDAR-PTQ.

Cons:

1. The adaptive foreground recognition and gradient-based weight sensitivity are based on empirical observations of existing 3D detectors. However, no failure cases are presented across detection or semantic segmentation tasks. In-depth analysis under challenging conditions such as very sparse, noisy, or occluded point clouds are required to studied.

2. The impact of the calibration dataset (e.g., data split, size, and statistical variance) is underexplored. This raises concerns about generalization under real-world deployment, especially when facing unseen domain shifts.

3. Experiments on pillar-based detectors like PointPillars and PillarNeXt would make the results more convincing, as these models already achieve high speed and BEV efficiency.

---

> ### Author Rebuttal · Authors · 2025-07-31
>
> We sincerely appreciate your constructive feedback. Below, we provide point-by-point responses, with all revisions incorporated accordingly to improve the quality and clarity of our manuscript.
>
> >**Q1: Failure cases and analysis under challenging conditions (e.g., sparse, noisy, or occluded point clouds).**
>
> **A1:** Thank you for your valuable suggestion. In response, we selected the sparse scenes as examples and performed new experiments to evaluate our quantization method under sparse scenarios (points with distances >20 m). All quantized models and baselines were re-evaluated under this setting.
>
> 1. **Quantization Performance in Sparse Scenarios**: As shown in Tab.1, our Point4Bit maintains robust performance even at low-bit widths setting (e.g., W4A4), achieving 19.88\% mAP and 43.76\% NDS, which are comparable with the FP model (20.50\% mAP, 44.85\% NDS). This demonstrates the robustness of our approach in sparse scenarios.
>
> 2. **Ablation of Core Modules in Sparse Scenarios**:  We also conducted ablation studies to verify that both G-KWQ and FA-PAQ modules consistently contribute to the overall accuracy and robustness of Point4Bit under these challenging sparse scenes (Tab.2). This indicates that our method's quantization performance is not coincidental but is a direct result of its design, which effectively identifies and preserves critical information even in degraded scenarios.
>
> We hope these additional results and the in-depth analysis further demonstrate the effectiveness of Point4Bit in challenging scenes.
>
>
> **Tab.1. Quantization results on nuScenes *val* set (CP-Voxel, sparse scenes).**
>
> | Methods      | Bits(W/A) | mAP   | NDS   | Car   | Truck | C.V. | Bus   | Trailer | Barrier | Motor. | Bike  | Ped.  | T.C.  |
> |--------------|-----------|-------|-------|-------|-------|------|-------|---------|---------|--------|-------|-------|-------|
> | Full Prec.   | 32/32     | 20.50 | 44.85 | 46.27 | 30.38 | 4.90 | 37.24 | 18.65   | 13.89   | 10.95  | 1.31  | 36.26 | 5.13  |
> | RTN          | 8/8       | 20.30 | 44.72 | 46.18 | 30.23 | 4.48 | 37.14 | 18.36   | 13.72   | 10.33  | 1.20  | 36.24 | 5.09  |
> | RTN+GS       | 8/8       | 20.48 | 44.81 | 46.25 | 30.35 | 4.94 | **37.40** | 18.58   | 13.85   | 10.82  | 1.29  | 36.20 | 5.10  |
> | **Ours**     | 8/8       | **20.54** | **44.84** | **46.28** | **30.38** | **4.99** | 37.14 | **18.63** | **13.98** | **11.24** | **1.33** | **36.26** | **5.14** |
> | RTN          | 4/8       | 19.11 | 43.69 | 43.66 | 28.03 | 6.44 | 33.88 | 17.67   | 12.65   | 8.79   | 0.46  | 34.62 | 4.80  |
> | RTN+GS       | 4/8       | 19.36 | 43.84 | 45.17 | 29.01 | 3.14 | 34.96 | **17.75** | 12.18   | 9.80   | **1.22**  | **35.84** | 4.51  |
> | **Ours**     | 4/8       | **20.03** | **44.21** | **45.19** | **29.23** | 5.94 | **35.48** | 17.61   | **13.21** | **11.67** | 1.20  | 35.68 | **5.01** |
> | RTN          | 4/4       | 6.96  | 31.90 | 20.09 | 11.50 | 0.00 | 14.41 | 2.28    | 1.76    | 0.00   | 0.00  | 19.20 | 0.31  |
> | RTN+GS       | 4/4       | 13.29 | 38.77 | 38.66 | 19.72 | 2.89 | 22.37 | 8.36    | 10.85   | 1.53   | 0.00  | 26.71 | 1.78  |
> | **Ours**     | 4/4       | **19.88** | **43.76** | **44.91** | **28.98** | **6.37** | **35.13** | **16.72** | **13.00** | **11.94** | **1.22** | **35.59** | **4.88** |
>
>
> **Tab.2. Ablation study of quantization components (CP-Voxel, sparse scenes).**
>
> | Methods | Bits(W/A) | G-KWQ | FA-PAQ | mAP   | NDS   |
> |---------|-----------|-------|--------|-------|-------|
> | Full Prec. | 32/32     | -     | -      | 20.50 | 44.85 |
> | Ours       | 4/4       | ✓     | -      | 15.61 | 40.59 |
> | Ours       | 4/4       | -     | ✓      | 17.93 | 42.11 |
> | Ours       | 4/4       | ✓     | ✓      | **19.88** | **43.76** |
>
>
> >**Q2: The impact of calibration dataset choices (e.g., split, size, variance).**
>
> **A2:** Thanks for your question.
> 1. Our quantization data selection follows established practices in previous works (e.g., PD-Quant[17], LiDAR-PTQ[38], and Quantization White Paper[23]), we use only a small fraction ~1% of the training data for calibration.
>
> 2. Actually, the calibration dataset has a minor impact in our method. To thoroughly investigate this, we performed ablation studies on the calibration set size, a key factor influencing its statistical representativeness. We evaluated our method on both detection (nuScenes/CP-Voxel) and segmentation (SemanticKITTI/LargeKernel3D) tasks, varing the calibration data size over a wide range (from 32 to 1,024 samples). As shown in Tab.3, our key finding is that the final quantized performance is remarkably stable and insensitive to the size of the calibration set. For example, on SemanticKITTI, the mIoU for W8A8 quantization remains stable between 70.1 and 70.3, and on nuScenes, the mAP for W8A8 varies minimally from 58.44 to 58.48. These results are consistent with prior work, and indicate that our method is not overfitting to the specific statistics of any particular data split and is insensitive to the choice of calibration data size.
>
> Overall, our experiments indicate that our quantization strategy generalizes well even with a small calibration set—an important property for real-world deployment. The results will update in final version.
>
> **References:**
> [17] PD-quant: Post-training quantization based on prediction difference metric.
> [23] A white paper on neural network quantization.
> [38] LiDAR-PTQ: Post-Training Quantization for Point Cloud 3D Object Detection.
>
> **Tab.3. Ablation study on the effect of calibration dataset size for nuScenes *train* set and SemanticKITTI *train* set. CD Size denotes the number of samples used for calibration.**
>
> | Bits(W/A) | CD Size | CP-Voxel mAP  | CP-Voxel NDS  | LargeKernel3D mIoU |
> |-----------|---------|------|------|------|
> | Full Prec | -       | 60.53| 66.64| 70.3 |
> | W8A8      | 64      | 58.44| 66.19| 70.1 |
> |           | 128     | 58.45| 66.20| 70.2 |
> |           | 256     | **58.48**| **66.21**| **70.3** |
> |           | 1024    | 58.44| 66.20| 70.2 |
> | W4A4      | 64      | 56.83| **64.96**| 70.1 |
> |           | 128     | 56.85| 64.84| 69.9 |
> |           | 256     | **56.97**| 64.88| 70.0 |
> |           | 1024    | 56.55| 64.68| **70.3** |
>
> >**Q3: Experiments on pillar-based detectors (e.g., PointPillars or PillarNeXt).**
>
> **A3:** Following your advice, we evaluated our Point4Bit method on the advanced pillar-based PillarNeXt detector, using the official pre-trained weights as the full-precision (FP) baseline.
>
> As shown in Tab.4, **our quantization results on PillarNeXt-B are consistent with those on voxel-based detectors** (Tab. 1 in the main paper). Under the W8A8 setting, our method achieves 62.44% mAP and 68.54% NDS, nearly matching the FP baseline (62.51% mAP, 68.61% NDS). Even in lower-bit scenarios (W4A8, W4A4), accuracy remains well preserved with only minimal degradation.
>
> These results demonstrate the generality and robustness of our approach across both voxel-based and pillar-based 3D detectors. All PillarNeXt results will be included in the final version of the paper.
>
> **Tab.4. Quantization results on nuScenes *val* set with PillarNeXt-B:**
>
> | Methods     | Bits(W/A) | mAP   | NDS   | Car   | Truck | C.V. | Bus   | Trailer | Barrier | Motor. | Bike  | Ped.  | T.C.  |
> |-------------|-----------|-------|-------|-------|-------|------|-------|---------|---------|--------|-------|-------|-------|
> | Full Prec.  | 32/32     | 62.51 | 68.61 | 84.77 | 58.60 | 21.44| 66.53 | 35.24   | 69.78   | 68.01  | 56.43 | 87.22 | 76.99 |
> | RTN         | 8/8       | 61.65 | 68.11 | 83.65 | 57.10 | 20.73| 66.36 | 32.79   | 69.44   | 67.33  | 55.53 | 87.03 | 76.48 |
> | RTN+GS      | 8/8       | 62.37 | 68.49 | 84.45 | 58.34 | 20.88| **66.51** | **35.43**   | **69.90**   | 67.89  | **56.28** | 87.04 | 76.96 |
> | **Ours**    | 8/8       | **62.44** | **68.54** | **84.76** | **58.55** | **21.18** | 66.49 | 35.27 | 69.63 | **68.02** | 56.21 | **87.20** | **77.01** |
> | RTN         | 4/8       | 59.66 | 66.00 | 83.10 | 56.12 | 18.53| 63.22 | 31.54   | 66.37   | 64.21  | 51.29 | 86.47 | 75.73 |
> | RTN+GS      | 4/8       | 60.40 | 66.85 | 83.58 | 57.04 | **19.94**| 62.64 | 32.98   | 67.80   | 65.46  | 53.52 | 85.84 | 75.15 |
> | **Ours**    | 4/8       | **61.27** | **67.19** | **84.44** | **57.58** | 19.54 | **63.97** | **35.15** | **68.36** | **66.92** | **54.28** | **86.54** | **75.92** |
> | RTN         | 4/4       | 9.31  | 25.49 | 21.61 | 7.99  | 0.00 | 9.26  | 3.81    | 0.11    | 1.60   | 0.00  | 38.45 | 10.26 |
> | RTN+GS      | 4/4       | 36.20 | 51.62 | 73.57 | 40.43 | 6.50 | 35.31 | 3.63    | 60.20   | 30.69  | 22.81 | 22.81 | 28.57 |
> | **Ours**    | 4/4       | **60.89** | **66.93** | **84.31** | **57.72** | **19.14** | **63.49** | **35.26** | **68.08** | **66.59** | **52.95** | **86.22** | **75.13** |

---

> > ### Comment · Reviewer_NnTp · 2025-08-04
> >
> > Thanks for the authors' reply. They have effectively addressed my concerns regarding the effectiveness of the proposed quantization techniques, particularly under challenging conditions (>20 meters) and on the stronger baseline, PillarNeXt. I will increase my rating to borderline accept.

---

> ### Author Response · Authors · 2025-08-04
> **Thanks for your recognition**
>
> Thank you for your positive feedback and for indicating that you would ​raise the score to borderline accept. We're glad our responses have effectively addressed your concerns. All the revised experiments (including results on PillarNeXt and challenging conditions) will be included in the final version. We will keep moving to make Point4bit and the following projects better and better.

---

> > ### Author Response · Authors · 2025-08-06
> >
> > Dear Reviewer NnTp,
> >
> > Thank you again for your positive feedback and for considering raising your score based on our rebuttal. As the discussion period is concluding soon, we just wanted to gently and respectfully follow up. We would be very grateful if you could submit your final rating at your convenience.
> >
> > We understand this is an incredibly busy time, and your support and time mean a lot to us. We are truly grateful for your valuable comments throughout the review process. Thank you once again!
> >
> > Sincerely,
> > The Authors of Submission 9212

---

### Official Review · Reviewer_ETfE · 2025-07-03

**Clarity:** 3
**Significance:** 3
**Originality:** 3
**Rating:** 4
**Confidence:** 3

**Summary:**

This paper presents Point4Bit, a post-training quantization (PTQ) framework designed to enable efficient 4-bit quantization for voxel-based 3D object detection models. Unlike prior PTQ approaches that are typically limited to 8-bit quantization and lack generalization across tasks, Point4Bit introduces two key techniques to address the challenges of low-bit quantization:

(1) Foreground-aware Piecewise Activation Quantization (FA-PAQ), which identifies semantically important foreground regions and applies finer-grained quantization to preserve critical features, and (2) Gradient-guided Key Weight Quantization (G-KWQ), which uses gradient sensitivity to retain important weights and reduce error propagation.

Experiments across multiple datasets and tasks demonstrate that Point4Bit achieves strong performance under 4-bit settings with minimal accuracy loss and significantly faster quantization time compared to existing methods.

**Questions:**

See weaknesses
I look forward to the authors’ rebuttal, which may influence the final score depending on how well the raised questions are addressed.

**Ethical Concerns:**

["NO or VERY MINOR ethics concerns only"]

**Limitations:**

yes

**Quality:**

3

**Strengths And Weaknesses:**

**Strengths**:
- The paper is clearly written and logically organized. It provides a strong motivation for enabling 4-bit post-training quantization in 3D point cloud detection to facilitate edge deployment.
- The proposed Point4Bit framework introduces two novel techniques: Foreground-aware Piecewise Activation Quantization (FA-PAQ) and Gradient-guided Key Weight Quantization (G-KWQ). Both are well-motivated and directly address core challenges in low-bit quantization.
- The method is fully post-training and requires no retraining or labeled data. It is evaluated across multiple 3D tasks including detection, classification, and segmentation, demonstrating strong generalization.
- The experimental results are comprehensive. The paper reports performance under multiple bit-width settings, ablation studies, comparisons with existing methods, and shows clear improvements in both accuracy and quantization efficiency.

**Weaknesses**:
- The paper does not report actual inference latency or throughput on hardware platforms such as NVIDIA Orin or Jetson. Given that efficient deployment is a central motivation, the absence of real-world performance measurements limits the evaluation of practical utility.
- Given the rising popularity of attention-based backbones for point cloud understanding, such as PTv3[1], it would be valuable to include evaluation on one of these architectures. This would help assess whether the proposed quantization strategy is compatible with transformer-based models beyond voxel and MLP designs.
- The classification results are reported on ModelNet40, a clean and synthetic dataset that lacks real-world complexity. While commonly used, the dataset is heavily saturated, with many methods achieving over 93% accuracy. More importantly, performance differences are often within 0.1–0.3%, making it difficult to determine whether improvements are statistically significant. In many cases, variation due to randomness (e.g.weight initialization, sampling) can outweigh the gains from the quantization method itself. Including results on more challenging datasets like ScanObjectNN[2] would provide a more reliable and robust evaluation.

[1] Point Transformer V3: Simpler, Faster, Stronger.

[2] Revisiting Point Cloud Classification: A New Benchmark Dataset and Classification Model on Real-World Data.

---

> ### Author Rebuttal · Authors · 2025-07-31
>
> We sincerely appreciate your constructive feedback. Below, we provide point-by-point responses, with all revisions incorporated accordingly to improve the quality and clarity of our manuscript.
>
> >**Q1: Inference latency on real hardware platforms (NVIDIA Orin or Jetson).**
>
> **A1:** Thanks for your question about inference latency. In our initial submission, we reported inference latency in Appendix Tab.1. In detail, the quantized CP-Voxel model achieves 31.1 FPS on Jetson AGX Orin (a 3× speedup over FP32) and 5.2 FPS on Jetson Xavier NX (a 2.7× speedup over FP32). The results demonstrate the practicality of our approach on widely used edge devices. We will report the inference results in the main paper to highlight the effectiveness of our quantization method in the final version.
>
> >**Q2: Evaluation on attention-based backbones (e.g., PTv3).**
>
> **A2:** Thank you for your suggestion. We agree that evaluating our method on transformer-based backbones, such as the excellent work PTv3 [42], is valuable given their rising popularity in point cloud understanding and perception. Our current work is intentionally focused on CNN-based point cloud models, primarily because transformer and CNN architectures present fundamentally different challenges from a quantization perspective, necessitating distinct optimization strategies.
>
> Quantizing transformer-based architectures like PTv3 has unique issues:
>
> 1. **Extreme Activation Outliers:** The self-attention mechanism (with its Softmax function) and extensive residual connections, are known to produce activation values with extreme outliers. These outliers significantly widen the dynamic range, which poses a major challenge for standard Post-Training Quantization (PTQ) and can lead to severe accuracy degradation.
> 2. **Different Normalization Layers:** Transformers typically employ Layer Normalization, whereas CNNs predominantly use Batch Normalization. The statistical properties and the effects these layers have on data distributions are entirely different. Consequently, many quantization techniques designed for CNNs, especially those that leverage Batch Normalization statistics, are not directly transferable to transformer models.
>
> Due to the above differences, the research community has developed separate streams of specialized quantization methods. For instance, as you noted, methods like PTQ4ViT [42] and FQ-ViT [43] are specifically tailored to address the outlier issues in Vision Transformers, while methods such as PD-Quant [17] and Q-Drop [31] are developed for CNNs.
>
> **In this paper, our intention is to push the bounds of quantization bit-width for widely-used CNN-based point cloud models; for transformer-based models such as PTv3 is beyond our initial scope.** We agree that developing quantization for transformers is a promising direction and a crucial next step, and we plan to explore this in future work.
>
> **References:**
> [17] PD-quant: Post-training quantization based on prediction difference metric.
> [31] QDrop: Randomly Dropping Quantization for Extremely Low-bit Post-Training Quantization.
> [42] Point Transformer V3: Simpler, Faster, Stronger.
> [43] PTQ4ViT: Post-Training Quantization for Vision Transformers with Twin Uniform Quantization.
> [44] FQ-ViT: Post-Training Quantization for Fully Quantized Vision Transformer.
>
> >**Q3: Results on more challenging datasets (e.g., ScanObjectNN) are needed for reliable and robust evaluation beyond ModelNet40.**
>
> **A3:** Thank you for your suggestion. Following your advice, we conducted experiments on the more diverse and challenging ScanObjectNN dataset. We report the quantization results of PointNet++ and PointNeXt.
>
> As shown in Tab.1 below, **our method's strong performance on the challenging ScanObjectNN dataset** confirms its advantages generalize effectively beyond synthetic data to realistic scenarios with noise and occlusions. Furthermore, the consistent performance trends across both datasets provide strong evidence that our method's improvements are robust and statistically significant. The results will update in final version.
>
> **Tab.1: Quantization results on the ScanObjectNN *val* set.**
>
> | Methods    | Bits (W/A) | PointNet++ OA | PointNet++ mAcc | PointNeXt OA | PointNeXt mAcc |
> |:-----------|:-----------|:-------------:|:---------------:|:------------:|:--------------:|
> | Full Prec. | 32/32      | 86.16         | 84.36           | 88.20        | 86.84          |
> | RTN        | 8/8        | 85.63         | 83.87           | 87.99        | 86.51          |
> | RTN+GS     | 8/8        | 85.74         | 83.94           | 87.99        | 86.57          |
> | **Ours**   | 8/8        | **86.02**     | **84.30**       | **88.20**    | **86.87**      |
> | RTN        | 4/8        | 82.79         | 80.03           | 86.16        | 84.59          |
> | RTN+GS     | 4/8        | 82.72         | 80.28           | 86.43        | 84.84          |
> | **Ours**   | 4/8        | **85.67**     | **83.71**       | **88.13**    | **86.65**      |
> | RTN        | 4/4        | 81.02         | 78.49           | 77.34        | 74.41          |
> | RTN+GS     | 4/4        | 80.99         | 77.97           | 78.28        | 78.28          |
> | **Ours**   | 4/4        | **85.46**     | **83.38**       | **87.47**    | **86.07**      |

---

### Comment · Area_Chair_YkV4 · 2025-08-04
**[Action Required] Participate in Reviewer-Author Discussion**

Dear Reviewers,

As the author-reviewer discussion period will end soon (Aug 6, 11:59 PM AoE), please take a moment to read the authors’ responses and post a reply - either to acknowledge their clarifications or to raise any remaining concerns.

Thank you for your time and contributions to the review process.

Best regards,

AC

---

### Note · Authors · 2025-08-14

Dear Program Chairs, Senior Area Chairs, Area Chairs, and Reviewers,

Thank you for your time and effort in handling our paper. We sincerely appreciate all reviewers’ constructive feedback, have thoroughly addressed all comments in our rebuttal, and integrated the corresponding revisions into the manuscript. During rebuttal, Reviewer NnTp, hG7x, and Uao9 all acknowledged that our responses solved their concerns. **Reviewer NnTp and Reviewer Uao9 increased their scores—NnTp to "borderline accept", Uao9 to "accept", and all reviewers ultimately gave consistent positive ratings for our submission.**

---

### Responses to Reviewer ETfE

Reviewer ETfE gave a **positive rating** and emphasized inference latency and dataset robustness. We addressed these by:
- **Reporting inference speedups** on Jetson AGX Orin/Xavier NX in the main paper, showing substantial gains over FP baselines.
- **Adding results on ScanObjectNN**, confirming strong robustness and deployment potential.

---

### Responses to Reviewer NnTp

Reviewer NnTp increased the rating to **"borderline accept"** after our rebuttal. Their main concerns were addressed by:
- **Adding experiments on sparse scenes**, showing robust performance in difficult settings.
- **Conducting ablations** to verify key module contributions.
- **Including results on PillarNeXt**, confirming general applicability.

---

### Responses to Reviewer hG7x

Reviewer hG7x gave a **positive rating** and focused on SOTA comparisons, clarity, and code release. We addressed these by:
- **Clarifying Point4Bit as the first strong 4-bit PTQ baseline** for point cloud models.
- **Revising figures and equations** for clarity.
- **Committing to open-source code release upon acceptance.**

---

### Responses to Reviewer Uao9

Reviewer Uao9 **increased the score to "accept"** after discussion. Their concerns were addressed by:
- **Clarifying our principled, importance-aware quantization strategy.**
- **Explaining limitations of previous 4-bit methods** and our improvements.
- **Highlighting pipeline efficiency and simplicity.**
- **Acknowledging domain sensitivity and future work.**
- **Providing more results across bit-widths and architectures.**

---

All concerns are fully resolved, with supplementary data validating Point4bit's stability, effectiveness, novelty, and applicability. Revisions are incorporated, and we trust these efforts reflect the robustness of our work.

Sincerely,
The Authors of Submission 9212

---

### Decision · Program_Chairs · 2025-09-17

**Decision:**

Accept (poster)

**Comment:**

All four reviewers finally gave borderline accept or accept scores (2 borderline accepts and 2 accpets). Overall, this paper introduces Point4Bit, a post-training 4-bit quantization method for 3D detectors, to enable deployment of these models on resource-constrained edge devices. Reviewers found the method novel and practical. Concerns around comparisons and design clarity were addressed in the rebuttal. Given the overall consensus and impact, the AC recommends acceptance.